# FRONTALK: BENCHMARKING FRONT-END DEVELOPMENT AS CONVERSATIONAL CODE GENERATION WITH MULTI-MODAL FEEDBACK

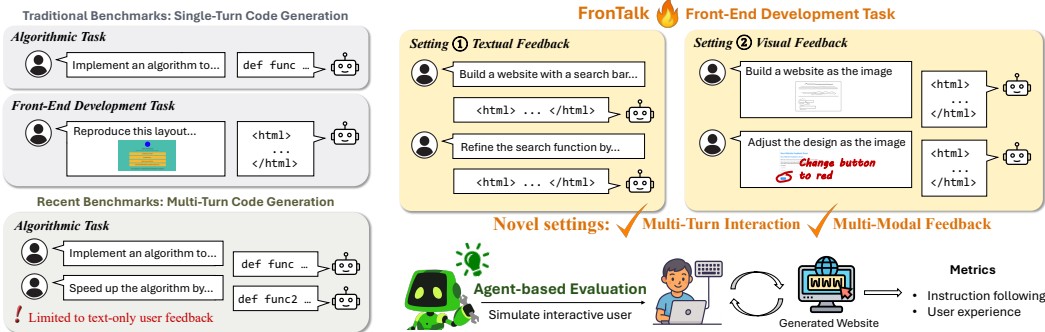

Figure 1: Focusing on the front-end development task, **FRONTALK** benchmark explores a novel setting with multi-turn interactions and multi-modal user feedback.

## ABSTRACT

We present **FRONTALK**, a benchmark for front-end code generation that pioneers the study of a unique interaction dynamic: **conversational code generation with multi-modal feedback**. In front-end development, visual artifacts such as sketches, mockups and annotated screenshots are essential for conveying design intent, yet their role in multi-turn code generation remains largely unexplored. To address this gap, we focus on the front-end development task and curate **FRONTALK**, a collection of 100 multi-turn dialogues derived from real-world websites across diverse domains such as news, finance, and art. Each turn features both a textual instruction and an equivalent visual instruction, each representing the same user intent. To comprehensively evaluate model performance, we propose a novel *agent-based evaluation framework* leveraging a web agent to simulate users and explore the website, and thus measuring both implementation correctness and user experience. Evaluation of 14 models reveals two key challenges underexplored in the literature: (1) a significant *forgetting issue* where models overwrite previously implemented features, resulting in task failures, and (2) a persistent challenge in *interpreting visual feedback*, especially for open-source vision-language models (VLMs). We propose a strong baseline to tackle the forgetting issue with ACECODER, a method that critiques the implementation of every past instruction using an autonomous web agent. This approach significantly reduces forgetting to **nearly zero** and improves the performance by up to **9.3%** (56.0%→65.3%). Overall, we aim to provide a solid foundation for future research in front-end development and the general interaction dynamics of multi-turn, multi-modal code generation.

## 1 INTRODUCTION

The success of Large Language Models (LLMs) in code generation has popularized the application of AI-assisted coding involving **multi-turn interaction**, as seen in trends like "vibe coding" and

a surge of commercial coding assistants. Reflecting this trend, recent benchmarking efforts have moved beyond single-turn code generation (Austin et al., 2021; Chen et al., 2021) to better measure how effectively LLMs respond to feedback across multiple turns (Wang et al.; Laban et al., 2025; Han et al.). However, these benchmarks predominantly focus on algorithmic or API-driven tasks, where multi-modal feedback is less central, and thus limit their evaluation to text-only interactions. This results in under-exploration of domains such as data visualization, application design, and front-end development where **multi-modal feedback** is a critical communication channel. For example, in front-end development, sketching a UI layout or annotating a screenshot is a natural and efficient method for communicating design changes. Therefore, this unique dynamic of multi-turn, multi-modal coding remains a significantly underexplored challenge.

To advance this frontier, we introduce FRONTALK, a benchmark for front-end development that integrates multi-turn code generation with multi-modal feedback. Our benchmark begins with a diverse and realistic collection of *user intents*, derived from real-world websites in the C4 dataset (Raffel et al., 2020) and spanning diverse domains such as e-commerce platforms, financial service sites, and digital art portfolios. To capture realistic conversational dynamics, we employ an LLM-based *user simulator* that generates context-aware instructions conditioned on prior dialogue. Crucially, to enable our simulated users to generate visual feedback, we equip them with a suite of *drawing tools* to communicate intents through sketching and annotating. The resulted FRONTALK dataset comprises 1,000 conversational turns across 100 dialogues, paired with 3,676 manually refined test cases, offering a robust foundation for evaluating multi-modal, multi-turn coding systems.

For evaluation, we propose an **agent-based evaluation framework** to evaluate both *instruction following* and *user experience*. For instruction following, we measure *pass rate* against human-annotated test cases using an automated web agent to interact with the website and verify task completion. This agent is uniquely equipped with *image manipulation tools* (e.g., cropping, image comparison), enhancing model's visual perception capabilities and improving alignment by 1.7% with human annotators on design tasks. For the nuanced dimension of user experience, we employ a *pairwise comparison* protocol. In this setup, first-time users simulated by LLMs interact with each website to learn the interface and attempt self-proposed tasks. A secondary LLM then compares the resulting trajectories and judges which interface is more *usable*, with criteria grounded in learning efficiency and satisfaction in task completion. Our evaluation protocol achieves a Cohen's Kappa of 0.63 for pass rate and 0.67 for usability, indicating significant alignment.

We evaluated a range of open- and closed-source models on our FRONTALK benchmark, including eight text-only LLMs evaluated on textual feedback and 12 VLMs evaluated on both textual and visual feedback. Our experimental results reveal several key findings:

1. **Open-source vs. proprietary gap**: Proprietary models, such as Gemini-2.5-Pro, maintain a substantial lead over open-source models. This disparity is particularly pronounced with visual feedback, where the performance gap is 24.1%, compared to the 12.5% gap with textual feedback. Since FRONTALK is the first to evaluate multi-turn multi-modal code generation, we attribute this gap to the superior generalization capabilities of proprietary models, an area where contemporary open-source VLMs still fall short.

2. **Forgetting issue:** The multi-turn setting introduces a significant forgetting issue, causing previously implemented functions or design elements to be contradicted or overwritten by later code. This issue consistently occurs across all leading models and results in a performance degradation of up to 46%, as measured by the *forgetting rate* metric.

3. **Challenges of visual feedback**: Visual feedback proves consistently more difficult than textual feedback, resulting in significantly lower pass rate. Common failure modes include (1) the literal replication of layout sketches without implementing the underlying functionality, and (2) overlooking textual annotations in densely annotated images.

4. **Key factors affecting usability**: The determinants of usability evolve with model capability. For weaker models, poor usability is primarily linked to *broken or non-functional features*; However, for stronger models that achieve high pass rates, usability is usually impacted by *design flaws*, such as inefficient navigation or a lack of interactive feedback. This highlights that achieving high usability requires implementing implicit, common-sense design principles that are often omitted from user instructions.

To address the critical forgetting issue, we propose ACECODER as a simple yet strong baseline using *a*gent *c*ritique to *e*nhance user instructions. For each turn in the conversation, ACECODER begins

with the LLM generating an *initial code solution* based on user instruction. Then, a web agent, powered by the same LLM, autonomously explore the rendered website to verify the implementation of both the current and past instructions. Coding failures or oversights identified by the agent are appended to the user instruction, prompting the model to explicitly prevent these specific pitfalls in the *secondary coding attempt*. Experiments show that ACECODER reduces forgetting rate to **nearly zero** and significantly improves the pass rate by up to **9.3%** with textual feedback. However, the performance gains are less significant with visual feedback, leading to gains up to **5.9%**, further highlights the challenge of effective code generation based on visual feedback.

In summary, our contributions are as follows: (1) We propose FRONTALK, a realistic benchmark for *multi-turn* front-end coding with both *textual* and *visual* feedback; (2) We identify critical limitations of current models, including significant performance degradation caused by *forgetting issue* and a significant difficulty in *interpreting visual feedback*, especially for open-source VLMs; (3) We propose ACECODER, an agent-based critique method that effectively mitigates the forgetting issue especially with textual feedback.

## 2 FRONTALK

### 2.1 TASK FORMULATION

**Data composition.** Each data point in our dataset represents a multi-turn dialogue, consisting of a sequence of user intents $\{\mathbf{i}_t\}_{t=1}^{T}$, where $T$ represents the number of turns[1]. To enable a granular evaluation of instruction following, each intent $\mathbf{i}$ is further paired with a set of test cases $\mathcal{C}(\mathbf{i}) = \{c_m\}_{m=1}^{M}$. These test cases decompose the paragraph-long intent into a series of individual, verifiable statements that form the basis for our evaluation. For example, the intent *implement visual badges next to thread titles(1) to indicate their status with distinct colors and shapes(2)* is broken down into test cases like: (1) A badge is displayed next to each thread title, and (2) Badges for different statuses are visually distinct.

**Inference pipeline.** The pipeline for evaluating a conversational model $\mathcal{M}(\cdot)$ is detailed in Alg. 1. At each turn $t$, model $\mathcal{M}$ produces an output code $o_t$ based on both the current instruction and the full conversation history $\mathbf{H}_t$. To capture the dynamic nature of multi-turn conversations, where later turns are conditioned on prior interactions, the static intent $\mathbf{i}$ is not used directly as model input. Instead, we employ a user simulator, $\text{SIMUSER}(\cdot)$, to transform the static intent $\mathbf{i}_t$ into a context-aware instruction $\tilde{\mathbf{i}}_t = \text{SIMUSER}(\mathbf{i}_t|\mathbf{H}_t)$, which will serve as the actual input to the model. The final website $w$, rendered from the last turn's output code $o_T$, is then used for evaluation.

---

**Algorithm 1** Inference pipeline.

**Require:** Model $\mathcal{M}$; user intents $\{\mathbf{i}_t\}_{t=1}^{T}$; user simulator $\text{SIMUSER}$
**Ensure:** website $w$
1: $\mathbf{H}_1 \leftarrow [\,]$
2: **for** $t \leftarrow 1$ **to** $T$ **do**
3:     $\tilde{\mathbf{i}}_t \leftarrow \text{SIMUSER}(\mathbf{i}_t|\mathbf{H}_t)$
4:     $o_t \leftarrow \mathcal{M}(\tilde{\mathbf{i}}_t|\mathbf{H}_t)$
5:     $\mathbf{H}_{t+1} \leftarrow \mathbf{H}_t \parallel \langle \tilde{\mathbf{i}}_t, o_t \rangle$
6: **end for**
7: $w = \text{RENDER}(o_T)$
8: **return** $w$

---

**Evaluation.** To comprehensively evaluate the quality of the generated websites, we assess two key dimensions: *instruction following* and *user experience*.

The evaluation of instruction following relies on the predefined set of test cases $\mathcal{C}(\mathbf{i})$. We define the **pass rate** (PR) as our primary metric for implementation correctness. This is derived by first calculating the pass count (PC) – the number of test cases passed for a given intent $\mathbf{i}$ – and then aggregating these counts across all intents for the final code output $o_T$:

$$\text{PC}(o|\mathbf{i}) = \sum_{c \in \mathcal{C}(\mathbf{i})} \mathbf{I}\left(\text{RENDER}(o) \text{ passes } c\right); \quad \text{PR}(o) = \frac{\sum_{\mathbf{i}} \text{PC}(o|\mathbf{i})}{|\sum_{\mathbf{i}} \mathcal{C}(\mathbf{i})|} \tag{1}$$

In practice, the pass count is measured by a web agent that interacts with the website and verifies each test case, as in §2.4. Motivated by observations that features implemented in earlier turns are often overwritten in later turns, we further introduce the **forgetting rate** (FR) metric. It measures

---

[1]Set to 10 in this work

how much functionality correctly implemented in prior turns ($t < T$) is lost in the final output:

$$\text{FR} = 1 - \frac{\sum_{t=1}^{T-1} \text{PC}(o_T|\mathbf{i}_t)}{\sum_{t=1}^{T-1} \text{PC}(o_t|\mathbf{i}_t)} \tag{2}$$

While these metrics assess implementation correctness when following user instructions, they may not fully capture the overall user experience. To address this, we also evaluate **usability**, defined as how easily a new user can learn to navigate the website and accomplish tasks. We evaluate this by deploying an LLM agent that simulates a first-time user with no prior knowledge of the website. The agent autonomously explores the interface, formulates tasks based on its observations, and attempts to complete them, yielding a behavioral trajectory. We then employ a *pairwise comparison protocol*, where an LLM judge reviews trajectories from two different websites and determines which provides a superior user experience. The final metric is the win rate against a curated set of reference websites, ensuring a robust and standardized evaluation.

**Single-turn baseline.** We also examine a single-turn baseline under an assumption that all user instructions are provided upfront before any interaction. Concretely, the model takes the concatenation of all user intents as input, formally denoted as $o = \mathcal{M}([\mathbf{i}_0, \ldots, \mathbf{i}_T])$. Note that this is *not* a directly comparable substitute for multi-turn interaction as it relies on an unrealistic assumption: In real-world dialogue, user requests naturally emerge in response to the evolving conversational context and cannot be faithfully reduced to a single, predetermined prompt. Instead, this baseline serves two **analytical purposes**: (1) It measures the intrinsic challenge of the user intents and thus benchmarks the inherent task difficulty by removing the complexity of multi-turn interaction; (2) It contributes to a broader research question: Is it easier for models to process all requirements at once, or incrementally with multiple reasoning steps? Comparing the performance between the single-turn and multi-turn settings yields mixed results, revealing distinct challenges for each setting, as discussed in §3.

**Further details.** Additional details are provided in subsequent sections: data curation in §2.2, user simulator in §2.3, and the agent-based evaluation in §2.4.

## 2.2 DATA CURATION

**Source website selection.** To curate a diverse and challenging set of user intents reflective of real-world scenarios, we begin by sampling 10,000 webpages from the C4 dataset (Raffel et al., 2020). For each site, we generate summaries of its content and key features using GPT-4o. We then apply BERTopic (Grootendorst, 2022) to these summaries to identify thematic clusters. From the most prominent clusters, we manually select 100 websites – one representative site per cluster – while excluding low-information pages (e.g., placeholders). This process yields a realistic and diverse set of seed websites spanning a wide range of domains, as summarized in Figure 2.

**Automatic data generation.** Using the curated website summaries, we prompt GPT-4o to generate user intents $\mathbf{i}_t$ and their paired test cases $\mathcal{C}(\mathbf{i}_t)$ for each conversational turn. These intents guide the subsequent user simulation (details in §2.3). We categorize user intents into two types: *functionality* (e.g., navigation, form submission, search) and *design* (e.g., layout, colors, images, visual effects). For each turn, we randomly select an intent type and prompte GPT-4o to generate a corresponding user intent along with a set of verifiable test cases. Figure 7 shows the distribution of words in the generated intents for each type, illustrating their diversity.

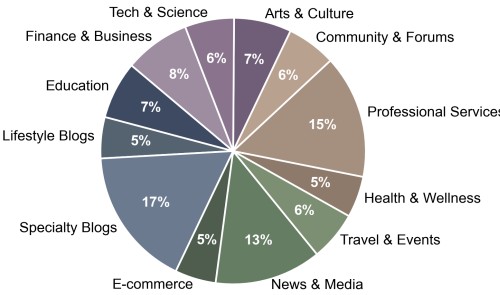

Figure 2: Website summary topics distribution.

| Dataset Size | |
|---|---|
| # Dialogues | 100 |
| # Turns | 1,000 |
| # Test Cases | 3,676 |
| *User Intent Length* | |
| # Words per Turn | 93.7 |
| # Words per Dialogue | 936.8 |

Table 1: Statistics of curated user intents and test cases in FRONTALK.

**User Intent** with 3 requirements, annotated by 1 2 3

① Integrate a "News and Updates" section... Ensure users can navigate to this section via a distinct navigation link labeled "News and Updates" on the homepage. ② Within this section, include a search bar that allows users to search for specific news articles or updates using keywords. ③ Additionally, implement a feature where users can subscribe to receive notifications for new updates directly from the "News and Updates" page.

**Textual Instructions** that *contextualize* user intent based on the website state

① First add a new navigation link labeled "News and Updates" **to the existing navigation bar at the top of the homepage**, ensuring it is distinct and easily accessible. **This section does not currently exist in the provided website code, so you will need to create a new section for it.** ② Within this new section, include a search bar that allows users to search for specific news articles or updates using keywords. ③ Additionally, provide a feature that enables users to subscribe to receive notifications for new updates directly from this page. **Ensure the design of this section is consistent with the existing style of the website, using the same color scheme and typography.**

**Visual Instructions** that *contextualize* user intent by annotating on website screenshots

Figure 3: **Illustration of the user simulators**. The textual simulator rewrites the user intent based on the current code state, while the visual simulator expresses the intent by drawing sketches or annotating screenshots of the website.

**Manual refinement and validation.** To ensure the reliability of our evaluation, we manually refined all automatically generated test cases to ensure that they are hallucination-free, unambiguous, and verifiable. We removed entirely unsatisfactory test cases and edited those that were partially satisfactory. This refinement process led to the removal of 66 test cases (1.7%) and the editing of 2,783 test cases (73.4%). To validate the improvement, a secondary independent annotator reviewed a sample of 224 test cases both before and after editing. The manual refinement increased the validity rate of test cases from 90.1% to **94.7%**. Statistics of the final dataset are in Table 1.

## 2.3 USER SIMULATION

The user intents are curated based on the assumption of an ideal website state. However, the actual website generated during a multi-turn conversation may deviate from this assumption, creating a contextual gap when intents are applied directly as model inputs. The objective of user simulation is to bridge this gap by dynamically adapting static user intents into context-aware *user instructions*, preserving the semantic core of the original intent while grounding it in the current state of the model-generated website. Specifically, user simulators perform two critical functions: (1) **Resolving ambiguity**: When an intent refers to an element that is ambiguous or absent on the current website, the simulator refines the reference (e.g., "the submit button" → "the submit button on the right") or requests the creation of the missing element. (2) **Omitting redundancy**: If an intent requests a feature that has already been implemented, the simulator adapts the instruction to avoid redundant requests (e.g., "implement a navigation bar" → "modify the existing navigation bar").

Following these principles, we implement two types of user simulators as in Figure 3:

1. **Textual instruction simulator**: An LLM is provided with the current code state and the original user intent, and is instructed to rewrite the intent into a textual user instruction.

2. **Visual instruction simulator**: A VLM is equipped with *drawing tools* and provided with screenshots of the current website. It produces sketches, shapes, and text annotations to anchor instructions directly onto the visual interface. This simulator is permitted multiple turns of drawing to fully articulate the user intent.

Implementation details and simulated instruction examples are presented in §D. We validate the reliability of our proposed user simulation with two metrics: (1) *intent preservation*, measured by the ratio of test cases well represented by the user instructions, and (2) *contextual faithfulness*, as measured by the ratio of user instructions that align well with the actual website state. A manual review of 114 simulated instructions and 411 test cases shows that textual simulators achieve high intent preservation (98%) and strong contextual faithfulness (96%), proving to be highly reliable. Visual simulators also exhibit high contextual faithfulness (97%) due to grounding user intents directly in rendered screenshots, but demonstrate much lower intent preservation (76%) owing to limited

tool-use effectiveness in GPT-4o. Despite this limitation, we view visual simulation as a valuable proof of concept: with our benchmark and more capable future models, the reliability of visual user simulation can be further improved.

### 2.4 Agent-Based Evaluation

Existing evaluation approaches in front-end development largely rely on static artifacts like rendered screenshots or source code (Si et al.; Sun et al., 2025). However, both approaches have fundamental limitations. For example, consider a dynamic test case: *selecting an option from the dropdown menu dynamically updates the page*. Screenshot-based metrics are not capable of verifying such cases, since they capture only a single visual state and cannot track dynamic user interactions. Meanwhile, code-based analysis, while theoretically more comprehensive, often misaligns with the end-user experience and may overlook subtle logical errors. In this example, the dropdown's filtering logic might appear syntactically correct and structurally sound, leading a code-based evaluation to register a false positive. Yet, a subtle bug could prevent it from functioning as intended – an error that only becomes apparent during actual user interaction.

To overcome these shortcomings and better emulate authentic user assessment, we employ an **interactive web agent** that evaluates the website through direct interaction. Our agent-based evaluator operates in two modes for the two different metrics: for *pass rate*, it acts as an informed expert, using the full context to verify a test case, while for *usability*, it simulates a naive user, exploring the site and attempting self-proposed tasks. We adapt the agent framework from WebVoyager (He et al., 2024) but introduce a novel suite of *image manipulation tools* to enhance the agent's visual perception, as detailed in §D.3. By enabling the VLM to perform operations like cropping UI elements and comparing visual states, these tools improve alignment with human judgments, particularly on design-centric tasks involving fine-grained visual details, such as verifying a subtle hover effect in a local region.

### 3 Experiments

We evaluate 20 models from nine model families, spanning proprietary models like ⑤OpenAI's GPT, Ⓐ\Anthropic's Claude, ◆Google's Gemini, as well as open-source models like ∞Meta's Llama, ☋DeepSeek-R1 (Guo et al., 2025), ⑦Qwen (Team, 2025a;b), ⑤OpenAI's GPT-OSS, ⓩ GLM (Hong et al., 2025) and ∨Ovis (Lu et al., 2025a). For evaluation, we employ the web agent powered by GPT-4o to perform agent-based evaluation. We report *pass rate* (PR) and *usability* (UX) as main metrics and *forgetting rate* (FR) as a supplementary metric to measure the forgetting issue.

### 3.1 Main Results

The main results are shown in Table 2. Overall, **all models perform substantially below perfect accuracy** on our challenging FRONTALK dataset. Even the state-of-the-art model, Gemini-2.5-Pro, falls short of perfect performance by 25.0% with textual feedback and 31.3% with visual feedback. The gap is more significant for open-source models, where the top models achieve only 62.5% with textual instructions and 44.6% with visual instructions. Notably, **the gap between closed-source and open-source model is more pronounced with visual feedback**. Since many models are heavily tuned for code generation from text, but not for code generation from image, we attribute the superior performance of proprietary models to stronger generalization capabilities. Our findings thus identify poor generalization as a key shortcoming of current open-source VLMs and an important direction for future work.

**Forgetting issue.** Table 2 reveals a significant forgetting issue across all models, with their forgetting rate ranging from 4.3% to 44.6%. The forgetting issue typically occurs when instructions from multiple turns request modifications to the same component of code (e.g., adding different buttons to the same section). Although these instructions are not mutually exclusive, the model is at risk of overwriting the previous implementation instead of integrating the new feature. Mitigating this issue requires models to accurately recall past instructions and dynamically navigate the codebase to preserve existing functionalities while implementing new ones – a capability that proves challenging for current models.

| | | Single-Turn (T) | | Multi-Turn (T) | | | Multi-Turn (V) | | |
|---|---|---|---|---|---|---|---|---|---|
| | | PR↑ | UX↑ | PR↑ | FR↓ | UX↑ | PR↑ | FR↓ | UX↑ |
| *Proprietary VLMs* | | | | | | | | | |
| GPT-4o | - | 51.4 | 50.0 | 56.0 | 21.4 | 55.0 | 55.0↓3.2 | 8.0 | 52.5↓2.5 |
| Claude-4-Sonnet | - | **77.5** | **71.8** | 45.5 | 22.8 | 60.8 | 59.3↑13.8 | 10.6 | 64.8↑4.0 |
| Gemini-2.5-Pro | - | 57.2 | 56.8 | **75.0** | **4.5** | **71.0** | **68.7**↓6.3 | **6.2** | **73.8**↑2.8 |
| *Open-Source LLMs* | | | | | | | | | |
| Qwen3 | 8B | 54.3 | 45.8 | 59.9 | 13.9 | 57.8 | n/a | n/a | n/a |
| GPT-OSS | 20B | 64.3 | 68.5 | 61.6 | 15.8 | **71.5** | n/a | n/a | n/a |
| Qwen3-Coder | 30B | 54.1 | 49.8 | 61.0 | **8.6** | 54.3 | n/a | n/a | n/a |
| Llama-3.3 | 70B | 38.4 | 35.8 | 28.9 | 46.3 | 40.8 | n/a | n/a | n/a |
| GPT-OSS | 120B | **68.0** | **73.5** | 57.9 | 9.2 | 66.8 | n/a | n/a | n/a |
| Qwen3 | 235B | 62.9 | 57.8 | 59.0 | 20.6 | 61.3 | n/a | n/a | n/a |
| Qwen3-Coder | 480B | 67.0 | 64.0 | **62.5** | 15.4 | 62.3 | n/a | n/a | n/a |
| DeepSeek-R1 | 685B | 60.0 | 53.5 | 24.7 | 34.5 | 40.0 | n/a | n/a | n/a |
| *Open-Source VLMs* | | | | | | | | | |
| Qwen2.5-VL | 7B | 24.5 | 19.5 | 23.7 | 28.2 | 26.8 | 16.1↓7.6 | 39.3 | 17.0↓9.8 |
| GLM-4.1V-Thinking | 9B | 36.6 | 32.3 | 18.2 | 5.8 | 41.5 | 15.3↓2.9 | **4.3** | 32.0↓9.5 |
| Ovis2.5 | 9B | 51.5 | 49.3 | 35.3 | 12.2 | 34.3 | 27.0↓8.3 | 10.5 | 39.5↑5.2 |
| Gemma3 | 12B | 47.1 | 43.5 | 43.8 | 30.3 | 43.0 | 30.4↓13.4 | 22.1 | 38.5↓4.5 |
| Gemma3 | 27B | 36.7 | 37.2 | 37.2 | 42.1 | 39.5 | 35.5↓1.2 | 16.7 | **41.0**↑2.5 |
| Qwen3-VL | 30B | **54.6** | **50.0** | **54.1** | **4.7** | **50.3** | 39.1↓15.0 | 12.3 | 35.8↓14.5 |
| Qwen2.5-VL | 32B | 44.5 | 38.0 | 33.2 | 46.9 | 37.5 | 36.9↑3.7 | 25.7 | 36.5↓1.0 |
| Qwen2.5-VL | 72B | 43.0 | 33.8 | 52.8 | 17.8 | 46.5 | **44.6**↓8.2 | 15.4 | 37.3↓9.2 |
| GLM-4.5V | 108B | 24.4 | 23.5 | 35.2 | 9.8 | 40.5 | 7.9↓27.3 | 27.7 | 7.8↓32.7 |

Table 2: **Main evaluation results.** We report the pass rate (PR↑) and usability (UX↑) as main metrics, and forgetting rate (FR↓) to additionally quantify the forgetting issue. The best performance in each block is **bolded**. For model performance with visual feedback (V), we report their difference against the performance with textual feedback (T) represented as ↓ or ↑.

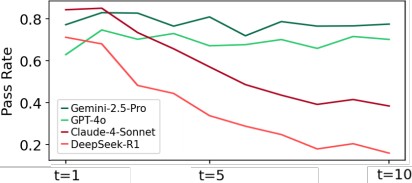

Figure 4: **Pass rate across different turns.** We show the pass rate of intermediate outputs, i.e. $PR(o_t|\mathbf{i}_t)$.

| | %Trunc. | PR↑ | UX↑ |
|---|---|---|---|
| Multi-turn, token limit = 30k | **5** | 42.0 | 52.1 |
| Multi-turn, token limit = 30k | 23 | 44.6 | 50.0 |
| Multi-turn, token limit = 30k | 88 | 47.8 | 54.2 |
| Single-Turn, token limit = 30k | 6 | **76.1** | **73.5** |

Table 3: **Claude-4-Sonnet performance when evaluated with different token limits for generation outputs.** For each setting, we report the ratio of truncated output (%Trunc.), pass rate (PR↑) and usability (UX↑).

**Long-context degradation.** Another key challenge is the extensive context length, which typically exceed 100k tokens. While models like GPT-4o and Gemini-2.5-Pro handle this length robustly, others suffer significant performance degradation. As shown in Figure 4, models such as Claude-4-Sonnet and DeepSeek-R1 achieve high pass rates in initial turns, but their performance drops sharply in later turns as context length grows. Table 3 shows an analysis on a subset of 12 dialogues (426 test cases), demonstrating that **reducing the context window improves multi-turn performance of Claude-4-Sonnet** – even with an extremely low token limit of 10k. However, all multi-turn settings still significantly underperform the single-turn baselines. These findings highlight long-context processing as a critical yet model-specific challenge, indicating a valuable direction for future work.

**Visual v.s. textual instructions.** As we evaluate VLMs with either textual or visual instructions, we are able compare their performance across the two settings. While the two instruction types are designed to convey the same set of user intents, Table 2 shows a consistent performance degradation for most VLMs when processing visual instructions, especially for open-source VLMs. To identify the challenges specific to visual instruction following, we manually analyze 608 test cases from the outputs of Qwen2.5-VL-72B and GPT-4o. Our analysis focus on the *subset* of

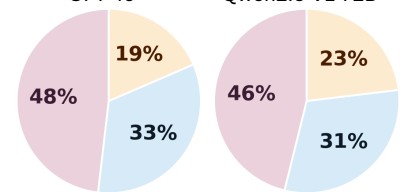

Figure 5: Error breakdown in interpreting visual instructions.

instances where a model fails a visual instruction but succeeds on its textual equivalent, thereby isolating errors caused by visual misinterpretation. We categorize these errors into three primary types, as shown in Figure 5: (1) **Failure to implement implicit functionalities**, the most common error, where VLMs replicate the UI layout but fail to implement the underlying functionalities (e.g., failing to implement the page-update behavior for navigation arrows, as in Figure 18); (2) **Missing text annotations**, where model miss some text annotations in the image, especially in densely anno­tated images (Figure 19); and (3) **Misinterpretation of visual clues**, the least common error, which occurs when visual clues are slightly ambiguous (Figure 20).

**Multi-turn v.s. single-turn.** As in §2.1, we study a single-turn baseline to (1) benchmark the inherent task difficulty, and (2) examine the broader research question of whether it is easier to solve a complex task in a single step or incrementally across multiple steps. Table 2 shows that the single-turn setting is highly challenging, with most models failing to exceed a 70% pass rate, confirming the intrinsic difficulty of our task. Analyzing the performance of the single-turn and multi-turn settings, we identify different challenges in each paradigm: while the multi-turn setting benefits from step-by-step task decomposition and reduced per-step complexity, it also suffers to forgetting and long-context degradation, as discussed above. Consequently, the overall picture is mixed and model-specific: some models (e.g., GPT-4o) perform better in multi-turn settings, while others (e.g., Qwen3-Coder-480B) achieve stronger results in single-turn settings.

## 3.2 SIMULATION OF AMBIGUOUS USERS

Prior sections assume that users express their intents through detailed, explicit instructions. In practice, however, users often provide indirect and ambiguous instructions – either because they have not yet fully clarified their own goals, or because they find it diffi-

| Turn | User | PR↑ | UX↑ |
|---|---|---|---|
| Single | - | 51.4 | 50.0 |
| Multi | Detailed (Standard) | **56.0** | **55.0** |
| Multi | Clarification-Only | 23.9 | 31.0 |
| Multi | Preference-Only | 16.4 | 21.0 |

Table 4: Performance of GPT-4o with users of different interaction types.

cult to articulate them without guidance. Consequently, LLMs must actively elicit or infer the user intents before executing tasks. To evaluate model performance in such ambiguous interactions, we simulate two user types: (1) **clarification-only user** that responds to direct questions but provides no unsolicited details, and (2) **preference-only user** that verbalizes only minimal requirements but instead reveals intent through choices among model-generated options. These simulated users have full access to the true intent but are constrained to communicate in the specified style. All settings share the same interaction budget ($N$ turns) to ensure fairness. As shown in Table 4, both user types significantly degrade performance compared to standard multi-turn or single-turn settings. This highlights models limited capability to actively elicit and infer user intent, directly affecting the task completion pass rate and the usability of generated websites.

## 3.3 ANALYSIS FOR AUTOMATIC EVALUATION

To validate the reliability of our automatic evaluation metrics, we recruit a team of five annotators, each having at least an undergraduate-level education in computer science, to conduct human eval­uation and measure the alignment between the automatic evaluation results and human judgments. We additionally measures the agreement across different LLMs as evaluators in Appendix E.1.

**Pass rate.** We tasked three annotators to independently evaluate 218 test cases. As in Table 5, our agent-based evaluator exhibits strong alignment with human judgments, achieving an accuracy of 80.6 and a Cohen's $\kappa$ of 59.2. We conduct an ablation study by removing our proposed im­age manipulation tools from the agent-based evaluator (w/o IM) and measure their resulted alignment scores. Results show that our full evaluator consistently achieves slightly higher alignment, with the benefit being more pronounced for design-category test cases. Figure 15 illustrates an ex­ample where these tools are critical for evaluating fine-grained visual details.

| | | Full | w/o IM |
|---|---|---|---|
| All Test Cases | $acc$ | 80.6 | 80.2↓0.4 |
| | $\kappa$ | 59.2 | 58.9↓0.3 |
| Design | $acc$ | 78.4 | 77.8↓0.6 |
| | $\kappa$ | 56.4 | 55.3↓1.1 |
| Functionality | $acc$ | 83.0 | 82.9↓0.1 |
| | $\kappa$ | 62.0 | 61.9↓0.1 |

Table 5: **Human alignment for the au­tomatic evaluation of pass rate**, mea­sured by agreement accuracy ($acc$) and Cohen's Kappa ($\kappa$). We compare our full evaluator against an ablated version with­out image manipulation tools (w/o IM).

**Usability.** We evaluate usability based on interaction trajectories generated by a web agent sim­ulating a first-time user. These trajectories exhibit plausible, human-like interaction behaviors, as

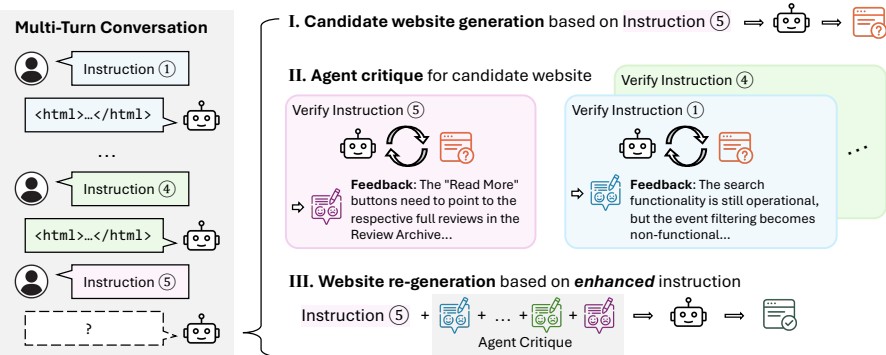

Figure 6: **Overview of ACECODER**. ACECODER first generates a candidate website, collects agent-based critique by interacting with the website and verifying the implementation of ***all prior instructions***, and then use the critique to enhance user instructions.

illustrated by the example in Table 14. To validate the reliability of automatic evaluation, we tasked three annotators to independently compare a total of 150 trajectory pairs. Analysis of human-LLM alignment yielded an accuracy of 69.3 and a Cohen's $\kappa$ of 56.3 (using quadratic weighting to handle tied votes), indicating moderate to substantial agreement. While the alignment is lower than the main metric of pass rate, reflecting the inherently subjective nature of user experience, it serves as an important complementary metric, filling a critical gap in user-centric evaluation for front-end development. Further analysis reveals that the determinants of usability shift with model proficiency: *For lower-performing models, usability is mostly affected by basic functionality*, where non-functional features like broken navigation links can severely degrade the user experience. *For higher-performing models, usability hinges on nuanced design choices* not always explicitly specified in user instructions, such as intuitive navigation (e.g., an easily accessible navigation bar) and clear system feedback (e.g., a confirmation message after a form submission). These findings highlight front-end development as a multi-dimensional challenge: Boosting user experience requires moving beyond mere implementation correctness to perform thoughtful, user-centric design – a critical direction for future research.

## 4 ACECODER: AN IMPROVED BASELINE TO ADDRESS FORGETTING

As discussed in §3.1, even state-of-the-art models suffer from the **forgetting issue**, where features implemented in prior turns are overwritten in later ones. To address this, ACECODER employs an **agent-based critique** process, detailed in Figure 6 and Algorithm 2. First, the model generates a candidate website. A web agent then interacts with this site to verify that all features – from both current and past instructions – are correctly implemented. The critiques gathered during this interaction are used to augment the original instructions. Finally, this enhanced prompt is fed back to the model to *regenerate* a final, improved website. We present implementation details in Appendix D.4. Though our critique agent adopts the same WebVoyager-based design as the evaluation agent, we eliminate the risk of label leakage through two measures: (1) the agent does not use the evaluation LLM, but instead uses the baseline LLM to which ACECODER is applied; and (2) the agent is given access only to the user instructions and the current code, and never to hidden test cases.

As in Table 6, ACECODER consistently improves performance across all evaluated models for both textual and visual instructions. The gains are particularly significant for textual instructions, reducing the forgetting rate to nearly zero. However, improvements for visual instructions are more modest, as the agent's ability to verify feature implementation is also bottlenecked by its capacity to interpret visual instructions.

We further conduct ablation studies to validate our design choices, as in Table 7. ACECODER outperforms the single-turn and multi-turn baselines, as well as an additional REPROMPT baseline that replays all prior instructions at each turn, as REPROMPT may fail to detect subtle implementation errors or complex code conflicts. We further perform three additional ablations: **(1) -Critique Past Turns**: A variant that only critiques the features from the current turn improves the pass rate but fails to meaningfully reduce the forgetting rate. This confirms that proactively verifying past features is critical to mitigating forgetting. **(2) -Agent Critique**: A variant that replaces the agent

| | Multi-Turn (T) | | | Multi-Turn (V) | | |
|---|---|---|---|---|---|---|
| | PR↑ | FR↓ | UX↑ | PR↑ | FR↓ | UX↑ |
| GPT-4o | 56.0 | 21.4 | 55.0 | 55.0 | 8.0 | 52.5 |
| +ACECODER | **65.3** | **0.4** | **76.3** | **60.2** | 7.2 | **65.8** |
| | +9.3 | -21.0 | +21.3 | +5.2 | -0.8 | +13.3 |
| Qwen2.5-VL-7B | 23.7 | 28.2 | 26.8 | 16.1 | 39.3 | 17.0 |
| +ACECODER | **25.4** | **-1.2** | **28.3** | **19.3** | 29.0 | **23.0** |
| | +1.7 | -29.4 | +1.5 | +3.2 | -10.3 | +6.0 |
| Qwen2.5-VL-72B | 52.8 | 17.8 | 46.5 | 44.6 | **15.4** | 37.3 |
| +ACECODER | **58.7** | **1.5** | **51.8** | **45.4** | 17.0 | **49.0** |
| | +5.9 | -16.3 | +5.3 | +0.8 | -1.6 | +11.7 |

Table 6: **Main results of ACECODER** with both textual (T) and visual (V) instructions.

| | PR↑ | FR↓ | UX↑ |
|---|---|---|---|
| *Baselines* | | | |
| Single-Turn | 51.4 | - | 50.0 |
| Multi-Turn | 56.0 | 21.4 | 55.0 |
| REPROMPT | 62.3 | 6.3 | 63.8 |
| *Ablations* | | | |
| **ACECODER** | **65.3** | 0.4 | **76.3** |
| - Critique Past Turns | 61.2 | 16.1 | 70.0 |
| - Agent Critique | 58.8 | **-0.8** | 66.3 |
| + Reflection | 63.4 | -0.2 | 76.0 |

Table 7: **Ablation study of ACECODER**, evaluated on GPT-4o with textual instructions.

critique with critique produced by a static LLM with access to the full conversation history, but not the built website. This results in a substantial performance degradation, highlighting the importance of interaction-based agent critique. **(3) +Reflection**: A variant that uses agent critiques to perform *reflection* rather than *regenerate* the entire website. While this variant also performs well, it is slightly inferior to our full regeneration-based approach, suggesting that complete regeneration offers greater flexibility for incorporating complex feedback.

## 5 RELATED WORK

**Front-end development.** Early research in front-end development focused on generating static webpages from images or textual descriptions (Gui et al., 2024; Laurençon et al., 2024; Si et al.; Xiao et al., 2025; Lin et al., 2025). Recent studies have begun to address interactivity by generating individual interactive components (Xiao et al., 2024) or entire navigable sites (Zhu et al., 2025; Lu et al., 2025c). However, they have predominantly considered a single-turn setting, while our work extends this line of research by introducing a multi-turn, conversational setting. The closest prior work is Li et al. (2025) that explores multi-turn code generation with visual inputs. However, they use visual input only in the first turn and is limited to low-fidelity low-informative sketches. In contrast, FRONTALK involves complex, realistic, and context-dependent visual inputs across all turns, as in Figure 8. A concurrent work (Lu et al., 2025b) also performs multi-turn front-end coding. However, they apply a multi-turn refinement with LLM-generated feedback as a *method* targeting a single-turn benchmark. In contrast, we formalize multi-turn front-end coding as a novel task and introduce a corresponding benchmark dataset.

**Multi-turn code generation.** Recent work in code generation has started to move beyond traditional single-turn benchmarks (Austin et al., 2021; Chen et al., 2021) toward multi-turn settings. These work simulates diverse conversational dynamics, such as users providing iterative feedback to help model complete coding tasks across multiple turns (Wang et al.; Han et al.), or users progressively clarifying ambiguous instructions (Laban et al., 2025). A common limitation, however, is their exclusive focus on textual interaction, overlooking the unique dynamics of visual feedback in multi-turn interactions. To mitigate this gap, our work incorporates visual feedback, which is critical for tasks like front-end development.

## 6 CONCLUSION

This paper introduces FRONTALK, a novel benchmark for front-end development that models a multi-turn code generation process with multi-modal user feedback. Our benchmark employs LLM-based user simulators to dynamically adapt pre-defined intents into either textual or visual instructions. The resulted websites are evaluated based on both instruction-following correctness and overall user experience. Evaluation of 14 models reveals that current models are far from perfect and struggle with two primary challenges: (1) a significant forgetting issue, where previously implemented features are overwritten in multi-turn interactions, and (2) a persistent difficulty in interpreting visual instructions. To address the former, we propose ACECODER, a strong baseline method that mitigates forgetting by leveraging a web agent to critique the implementation against past and present instructions. This work provides a solid foundation and a challenging testbed for future research in building more capable and reliable code generation approaches.

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

# Appendix

## A    ETHICS STATEMENT

In preparing this manuscript, we used large language models (LLMs) solely for the purpose of polishing the language (e.g., improving grammar and clarity). The LLMs were **not** employed to generate ideas, conduct analyses, provide interpretations, or alter the substance of the work. All conceptualization, methodology, data analysis, and interpretation were performed entirely by the authors. The use of the LLM was restricted to stylistic refinement, ensuring that the meaning and originality of the content remain unchanged.

## B    DATASET DETAILS

Wordcloud plots demonstrating distribution of user intents are in Figure 7.

**Functionality**
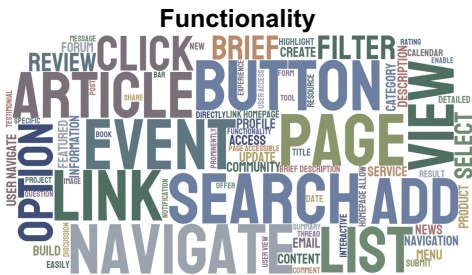
**Design**
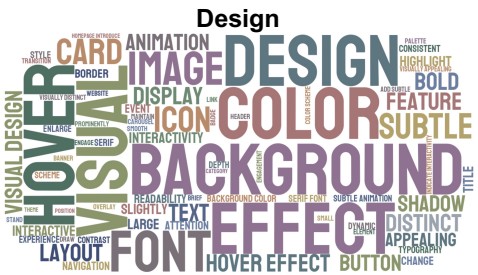

Figure 7: Distributions of user intents for functionality and design types respectively.

A data example is as follows:

- **Website goal:** The website "Shifting the Balance" appears to be a personal or family blog that shares thoughts and reflections on life, family, and societal issues.
- **Intent (t=1):** `[type=functionality]` Build a website for "Shifting the Balance" that serves as a personal blog sharing thoughts on life, family, and societal issues. Implement a comment section accessible at the bottom of each blog post page, allowing users to leave comments and replies. Include an email subscription feature on the homepage and each blog post page, enabling visitors to subscribe with their email addresses to receive notifications about new posts. Provide social media sharing options for Twitter, Facebook, and LinkedIn on each blog post page to facilitate easy sharing of content.
  - *Test cases:*
  - Navigate to the bottom of any blog post page to find the comment section. `[pass]` Users can see a text input field to write comment and an option to submit it. `[fail]` Users cannot write comment, or the comment cannot be submitted.
  - On the homepage and any blog post page, locate the email subscription feature. `[pass]` Users can see an input field to input email address, and an option to confirm subscription. `[fail]` The input field for email address is missing, or the email address cannot be submitted.
  - On any blog post page, find the social media sharing options. `[pass]` Users can see buttons or links for sharing on Twitter, Facebook, and LinkedIn. `[fail]` One or more of the social media sharing options is missing.
- **Intent (t=2):** `[type=functionality]` Enhance the comment section by adding a "Like" feature to each comment and reply, allowing users to express appreciation for comments. Ensure that each comment and reply has a visible "Like" button next to it, which increments a visible like counter when clicked. Users should be able to navigate to the comment section by scrolling to the bottom of any blog post page from the homepage.
  - *Test cases:*
  - Each comment and reply in the comment section of any blog post page has a visible "Like" button next to it. `[pass]` Every comment and reply displays a "Like" button adjacent to it. `[fail]` One or more comments or replies do not have a "Like" button next to them.
  - Clicking the "Like" button next to each comment or reply in the comment section at the bottom of any blog post page increases its like counter by one. `[pass]` The like counter for a comment

or reply increases by one when its "Like" button is clicked. `[fail]` The like counter does not change, or changes incorrectly, when the "Like" button is clicked.

- The like counter is displayed for each comment and reply in the comment section at the bottom of each blog post page. `[pass]` A like counter is visibly displayed for every comment and reply. `[fail]` The like counter is missing for one or more comments or replies.

- **Intent (t=3):** `[type=functionality]` Introduce a "Featured Posts" section on the homepage, showcasing a rotating selection of three highlighted blog posts to draw visitor attention. Ensure that users can navigate to the detailed blog post page by clicking on any featured post title or image. Include a brief excerpt and a placeholder image for each featured post to provide a preview of the content. The rotation should automatically change the featured posts every 10 seconds, but users should also have the option to manually navigate through them using forward and backward arrows.

  – *Test cases:*
  – The homepage displays a "Featured Posts" section with three distinct blog post previews. `[pass]` The homepage shows a section labeled "Featured Posts" with three previews of different blog posts. `[fail]` The "Featured Posts" section is missing, or it does not contain exactly three blog post previews.
  – Each featured post in the "Featured Posts" section on the homepage has a visible title, excerpt, and placeholder image. `[pass]` All three featured posts show a title, an excerpt, and a placeholder image. `[fail]` Any of the featured posts is missing a title, an excerpt, or a placeholder image.
  – Clicking on a featured post's title or image on the homepage navigates the user to the detailed blog post page. `[pass]` Clicking on the title or image of a featured post redirects the user to its detailed blog post page. `[fail]` Clicking on the title or image does not navigate to the detailed blog post page.
  – The featured posts on the homepage automatically rotate every 10 seconds. `[pass]` The set of featured posts changes automatically every 10 seconds. `[fail]` The featured posts do not change automatically, or the timing is incorrect.
  – Users can manually navigate through the featured posts on the homepage using forward and backward arrows. `[pass]` The user can click forward and backward arrows to manually navigate through the featured posts. `[fail]` The arrows are missing, or clicking them does not navigate through the featured posts.

- **Intent (t=4):** `[type=design]` Refine the visual design of the blog post pages by introducing a cohesive color scheme and typography style that enhances readability and aesthetic appeal. Use a serif font for blog titles and a sans-serif font for body text to create a clear hierarchy and visual contrast. Apply a soft background color to the entire page to reduce eye strain and highlight the content areas. Ensure that interactive elements such as social media sharing buttons, comment section, and "Like" buttons have a distinct hover effect to visually indicate interactivity.

  – *Test cases:*
  – Blog titles on each blog post page are displayed in a serif font, while body text uses a sans-serif font. `[pass]` Blog titles appear in a serif font, and body text is in a sans-serif font. `[fail]` Both blog titles and body text appear in the same font style.
  – The background color of each blog post page is soft and distinct from the content areas. `[pass]` The page background is a soft color, differing from the content sections. `[fail]` The page background color is harsh or indistinct from content areas.
  – Interactive elements such as social media sharing buttons, the comment section, and "Like" buttons on each blog post page exhibit a distinct visual change on hover. `[pass]` Interactive elements change visually (e.g., color, size) when hovered over. `[fail]` Interactive elements show no visual change when hovered over.

- **Intent (t=5):** `[type=design]` Enhance the homepage by designing an inviting hero section at the top, featuring a large, captivating background image that represents the blog's themes. Overlay this image with a semi-transparent color layer to ensure text readability. Include a welcoming headline in a bold serif font, accompanied by a concise tagline in a smaller sans-serif font. Add a call-to-action button, styled with a distinct hover effect, encouraging visitors to explore the blog further.

  – *Test cases:*

- The hero section on the homepage features a large background image. [pass] A large background image is prominently displayed at the top of the homepage. [fail] There is no background image, or it is not prominently displayed.
- The homepage's hero section should have a background image with a semi-transparent color overlay. [pass] The background image has a visible semi-transparent color layer over it. [fail] The background image does not have any overlay, or the overlay is not semi-transparent.
- The headline is in a bold serif font, and the tagline is in a smaller sans-serif font. [pass] The headline uses a bold serif font, and the tagline uses a smaller sans-serif font. [fail] The headline and tagline do not follow the specified typography styles.
- The call-to-action button on the homepage hero section is present with a distinct hover effect. [pass] A call-to-action button is visible and exhibits a distinct visual change when hovered over. [fail] There is no call-to-action button, or it lacks a hover effect.

- **Intent (t=6):** [type=design] Enhance the visual appeal of the "Featured Posts" section by adding smooth transition animations when rotating between posts, creating a more engaging user experience. Introduce a subtle zoom-in effect on the placeholder images of each featured post when hovered over, emphasizing interactivity and drawing attention. Ensure the navigation arrows have a bold, contrasting color to make them easily identifiable, and implement a gentle bounce animation on hover to indicate their functionality.

  - *Test cases:*
  - When the "Featured Posts" section on the homepage rotates between posts, there is a smooth transition animation observable. [pass] The transition between posts is smooth and visually fluid. [fail] The transition appears abrupt or jerky.
  - Hovering over a placeholder image in the "Featured Posts" section on the homepage results in a noticeable zoom-in effect. [pass] The image enlarges slightly when hovered over. [fail] The image remains static with no change on hover.
  - The navigation arrows in the "Featured Posts" section on the homepage are in a bold, contrasting color compared to the background. [pass] The arrows stand out distinctly against the background. [fail] The arrows blend into the background or are difficult to discern.
  - Hovering over the navigation arrows in the "Featured Posts" section on the homepage triggers a gentle bounce animation. [pass] The arrows exhibit a bounce effect when hovered over. [fail] The arrows show no movement or animation on hover.

- **Intent (t=7):** [type=functionality] Introduce a search functionality on the homepage, allowing users to easily find specific blog posts by entering keywords. Ensure the search bar is prominently positioned near the top of the homepage and includes a placeholder text prompting users to "Search blog posts...". Upon entering a search query and hitting enter, users should be directed to a results page displaying a list of blog posts that match the search terms. Each result should include the blog post title, a brief excerpt, and a link to the full post for easy navigation.

  - *Test cases:*
  - Verify the presence of a search bar near the top of the homepage with placeholder text "Search blog posts...". [pass] The search bar is visible near the top of the homepage with the specified placeholder text. [fail] The search bar is not visible, or the placeholder text is incorrect or missing.
  - Test the search functionality on the homepage by entering a keyword in the search bar and pressing enter. [pass] The user is directed to a results page displaying a list of blog posts matching the keyword. [fail] The user is not directed to a results page, or no relevant posts are displayed despite matching content.
  - Check that each search result on the results page includes the blog post title, a brief excerpt, and a link to the full post. [pass] Each result displays the title, excerpt, and a clickable link to the full post. [fail] Any of the required elements (title, excerpt, link) are missing from the search results.

- **Intent (t=8):** [type=functionality] Introduce a "Related Posts" section at the end of each blog post page, featuring three blog posts with similar themes to engage readers further. Ensure that each related post displays a title, a brief excerpt, and a placeholder image, with links to navigate to the full blog post. Allow users to access this section by scrolling to the bottom of a blog post page, just above the comment section. Incorporate a feature that dynamically selects related posts based on shared tags or categories with the current post.

- **Test cases:**
- Navigate to the bottom of each blog post page and verify the presence of a "Related Posts" section. `[pass]` The "Related Posts" section is visible just above the comment section. `[fail]` The "Related Posts" section is not visible or appears below the comment section.
- Check that each related post in the "Related Posts" section at the end of each blog post page displays a title, a brief excerpt, and a placeholder image. `[pass]` Each related post shows a title, an excerpt, and an image. `[fail]` One or more related posts are missing a title, an excerpt, or an image.
- For each post in the "Related Posts" section at the end of each blog post page, users can easily navigate to their respective full blog post. `[pass]` Users can navigate to the full blog post, such as by clicking the title, the image, or a clickable link. `[fail]` Clicking a related post title or image does not redirect to the corresponding full post page, and there are no visible navigation links or buttons.
- Verify that the related posts on each blog post page are selected based on shared tags or categories with the current post. `[pass]` The related posts share at least one tag or category with the current post. `[fail]` None of the related posts share any tags or categories with the current post.
- **Intent (t=9):** `[type=design]` Enhance the "Related Posts" section by incorporating a visually appealing card layout that neatly organizes each related post with a subtle drop shadow and rounded corners. Use a consistent background color across all cards to unify the section visually. Implement an interactive hover effect on each card, slightly elevating it to indicate interactivity and engagement. Ensure that the title of each related post is in a bold serif font for emphasis, while excerpts remain in a smaller sans-serif font for readability.
    - **Test cases:**
    - Each related post is displayed within a card with rounded corners and a subtle drop shadow on the blog post page. `[pass]` Each related post appears within a bordered card with rounded corners and a shadow effect. `[fail]` Related posts lack rounded corners or a shadow effect, or do not appear within a card layout.
    - All cards in the "Related Posts" section on each blog post page have a consistent background color. `[pass]` The background color is uniform across all related post cards. `[fail]` The background color varies between related post cards.
    - A hover effect elevates each card in the "Related Posts" section at the end of each blog post page slightly when interacted with. `[pass]` Hovering over a related post card causes it to slightly elevate from the rest. `[fail]` Hovering over a related post card does not change its position or appearance.
    - The title of each related post in the "Related Posts" section at the end of each blog post page is in a bold serif font, while excerpts are in a smaller sans-serif font. `[pass]` Titles are distinctly bold and serif, and excerpts are visibly smaller and sans-serif. `[fail]` Titles and excerpts do not follow the specified font styling, lacking contrast in weight or style.
- **Intent (t=10):** `[type=design]` Enhance the visual design of the email subscription feature across the homepage and each blog post page by introducing a stylish subscription form. Use a rounded input field with a placeholder text that invites users to "Enter your email...". Add a vibrant call-to-action button next to the input field, styled with a hover effect that changes its background color. Ensure the form is positioned prominently within the layout, and add a subtle animation that draws attention to the form upon page load.
    - **Test cases:**
    - The email subscription form on the homepage and each blog post page should feature a rounded input field with placeholder text "Enter your email...". `[pass]` The input field is rounded and displays the specified placeholder text. `[fail]` The input field is not rounded or does not display the specified placeholder text.
    - The call-to-action button next to the input field in the email subscription form should have a vibrant background color and change color on hover. `[pass]` The button has a vibrant background color and visibly changes color when hovered over. `[fail]` The button does not have a vibrant background color or does not change color on hover.
    - The email subscription form should be prominently positioned within the layout of the homepage and each blog post page. `[pass]` The form is easily noticeable and positioned in a prominent area of the page. `[fail]` The form is not easily noticeable or is positioned in a less prominent area of the page.

– The email subscription form on the homepage and each blog post page should feature a subtle animation upon page load to draw attention. [pass] A subtle animation is visible when the page loads, drawing attention to the form. [fail] No animation is present upon page load, or the animation does not draw attention to the form.

We further show the comparison between our proposed FRONTALK and the most related prior work, Sketch2Code (Li et al., 2025), in Figure 8.

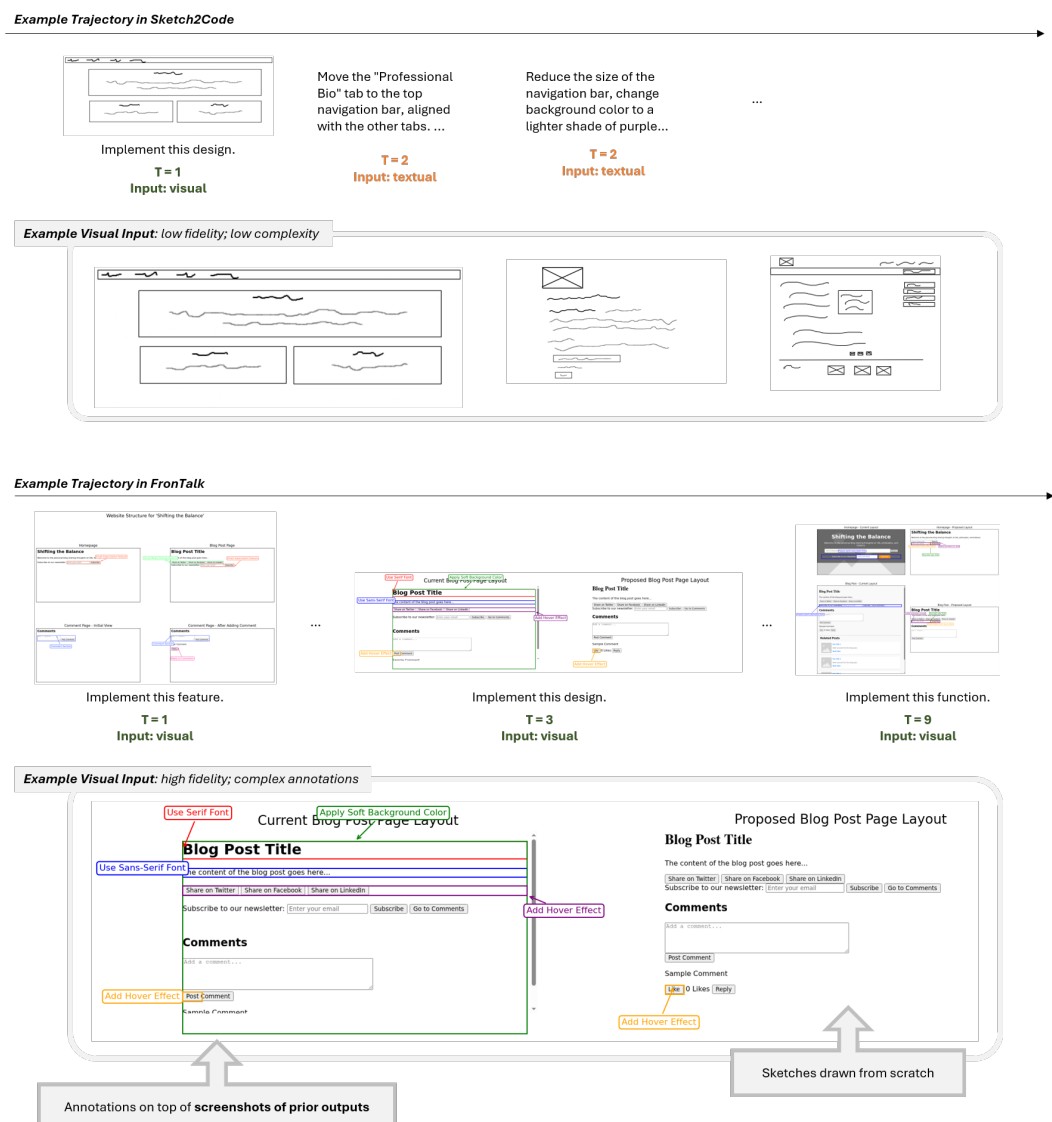

Figure 8: Comparison of FRONTALK and Sketch2Code (Li et al., 2025).

## C LLM INFERENCE DETAILS

### C.1 HYPERPARAMETERS

For LLM inference, we use the same configuration across every component, involving base inference on the benchmark, LLM-based evaluation, and the AceCoder pipeline. For open-source LLMs, we employ (Kwon et al., 2023) for model serving and make requests using the openai package; for proprietary LLMs, we directly call their official APIs.

We set the maximum output-token limit empirically for each model family to ensure that the model can complete its generation for more than 95% of input instructions. The limits are configured as follows:

- 10,000 tokens for GPT-4o, Llama-3.3, Qwen2.5-VL, GPT-OSS and Gemma3 models;
- 20,000 tokens for Gemini-2.5-Pro, Qwen3, Qwen3-Coder, DeepSeek-R1, GLM-4, Ovis2.5, and Qwen3-VL models;
- 30,000 tokens for Claude-4-Sonnet.

If the sum of input and output tokens exceeds the model's context window, we truncate the dialogue history by removing model responses from the earliest turns while always preserving the user instructions.

Except for the maximum token limit, we do not modify any additional hyperparameters. Each LLM is run using its default configuration – typically optimized for each model. For open-source models, we use the settings specified in their `generation_config.json`; for proprietary models, we rely on their default generation parameters. The Hugging Face identifiers for all open-source models are provided in Table 8.

| Model | Size | HF ID |
|---|---|---|
| Qwen3 | 8B | Qwen/Qwen3-8B |
| GPT-OSS | 20B | openai/gpt-oss-20b |
| Qwen3-Coder | 30B | Qwen/Qwen3-Coder-30B-A3B-Instruct |
| Llama-3.3 | 70B | meta-llama/Llama-3.3-70B-Instruct |
| GPT-OSS | 120B | openai/gpt-oss-120b |
| Qwen3 | 235B | Qwen/Qwen3-235B-A22B |
| Qwen3-Coder | 480B | Qwen/Qwen3-Coder-480B-A35B-Instruct-FP8 |
| DeepSeek-R1 | 685B | deepseek-ai/DeepSeek-R1-0528 |
| Qwen2.5-VL | 7B | Qwen/Qwen2.5-VL-7B-Instruct |
| GLM-4.1V-Thinking | 9B | zai-org/GLM-4.1V-9B-Thinking |
| Ovis2.5 | 9B | AIDC-AI/Ovis2.5-9B |
| Gemma3 | 12B | google/gemma-3-12b-it |
| Gemma3 | 27B | google/gemma-3-27b-it |
| Qwen3-VL | 30B | Qwen/Qwen3-VL-30B-A3B-Instruct |
| Qwen2.5-VL | 32B | Qwen/Qwen2.5-VL-32B-Instruct |
| Qwen2.5-VL | 72B | Qwen/Qwen2.5-VL-72B-Instruct |
| GLM-4.5V | 108B | zai-org/GLM-4.5V |

Table 8: Detailed `huggingface` ID for open-source models we evaluate.

## C.2 COST ANALYSIS

To measure the evaluation pipeline cost and facilitate wider community adoption, we measured the cost for each major component of our pipeline using GPT-4o on the full dataset. As shown in Table 9, the evaluation components based on web agents account for the largest share of cost, followed by the visual user simulation that relies on a tool-use drawing agent.

| | # LLM requests | # Input tokens | # Cached input tokens | # Output tokens | Est. cost |
|---|---|---|---|---|---|
| Inference (textual) | 1.0k | 147k | 7.2M | 2.0M | $29.43 |
| Inference (visual) | 1.0k | 1.2M | 9.9M | 1.3M | $27.95 |
| User simulation (textual) | 900 | 3.6M | 0 | 115k | $10.11 |
| User simulation (visual) | 2.0k | 16.1M | 12.7M | 1.0M | $66.38 |
| Evaluation of pass rate | 13.3k | 16.2M | 36.4M | 893k | $95.00 |
| Evaluation of usability | 1.6k | 2.2M | 3.8M | 258k | $12.75 |

Table 9: Cost breakdown for each component in the inference and evaluation pipeline.

We additionally quantify the additional computational overhead brought by our proposed ACE-CODER. We measure detailed breakdown of the inference cost using GPT-4o on a subset of 10 dialogues. As in Table 10, the majority of the overhead comes from agent-based critique.

| | # LLM requests | # Input tokens | # Cached input tokens | # Output tokens | Est. cost |
|---|---|---|---|---|---|
| Baseline inference | 100 | 14.4k | 728k | 205k | $3.00 |
| ACECODER | 4.5k | 3.1M | 15.1M | 967k | $36.22 |
| *Cost breakdown of* ACECODER *components* | | | | | |
| Initial website generation | 100 | 14.7k | 928k | 303k | $4.22 |
| Agent-based critique | 4.3k | 2.9M | 13.0M | 380k | $27.29 |
| Refined website generation | 90 | 148k | 1.2M | 284k | $4.7 |

Table 10: Cost analysis of ACECODER.

**The analysis above largely reflect the broader efficiency limitations of current LLM agents**, including web agents and tool-using agents, which involve multi-turn interactions with the environment, high-complexity observation spaces, and long multi-modal histories. These challenges are well known in existing systems and are not specific to our implementation. Potential future directions for improving efficiency include more efficient agent designs, better history-compression mechanisms, and improved planning algorithms.

# D    IMPLEMENTATION DETAILS

## D.1    TEXTUAL INSTRUCTION SIMULATOR

As described in §2.3, the textual instruction simulator is an LLM prompted with the current code state and asked to rewrite the user intent to (1) resolve ambiguity and (2) remove redundancy. The prompts for rewriting functionality- and design-type user intents are provided in Listing 6 and Listing 7, respectively. We prompt GPT-4o to simulate textual instructions. Example outputs for the user intents in §B are in Table 11.

## D.2    VISUAL INSTRUCTION SIMULATOR

For the visual instruction simulator, we equip it with drawing tools based on the `matplotlib` package. In the first turn, the model is provided with screenshots of every page on the website, as well as the coordinates of all visible elements. These coordinates can be used in later turns to guide annotations. At each turn, we provide a VLM (GPT-4o in this work) with the following tools:

1. Subplot organization: Tools to create and manage multiple subplots using `matplotlib`'s subplot functionality.
2. Layout visualization: Renders a layout sketch, represented by minimal HTML code, into an image. This process generates additional artifacts: the coordinates of every element on the new layout page. These coordinates are provided to the model in the next turn to assist with drawing.
3. Shape drawing: Tools for drawing various shapes using `matplotlib`.
4. Text annotation: A custom text annotation tool. We constrain the VLM to call our tool instead of `matplotlib`'s native text function. Our tool automatically organizes text annotations to minimize overlapping.

The VLM will generate python scripts to draw a visual instruction image using these tools. If the generated code has errors during execution, we allow models to re-generate for up to 5 times per turn. At the end of each tool-calling turn, the results, including the drawn images and other artifacts (e.g., coordinates), are fed back to the model. The model is then asked if the visualization is finished, i.e., if the visual instruction fully represents the user's intent. If not, it is allowed to perform another round of tool calling. We allow a maximum of three turns. The full prompt is in Listing 8.

Example outputs for the user intents from §B are shown in Figure 10-13. To demonstrate the multi-turn drawing process, we show the code snippets in two turns of instruction drawing in Listing 1 and Listing 2, with their corresponding output images in Figure 9 and 10.

| t | User Intents | Simulated Instructions |
|---|---|---|
| 2 | Enhance the comment section by adding a "Like" feature to each comment and reply, allowing users to express appreciation for comments. Ensure that each comment and reply has a visible "Like" button next to it, which increments a visible like counter when clicked. Users should be able to navigate to the comment section by scrolling to the bottom of any blog post page from the homepage. | To enhance user engagement in the comment section of each blog post, add a "Like" feature to both comments and replies. Implement a "Like" button next to each comment and reply, which should visibly increase the like counter upon being clicked. Ensure this functionality is available within the existing comment section located at the bottom of each blog post page. Users can access the comment section by scrolling down to the end of any blog post, which they can navigate to from the homepage by selecting a blog post title under "Latest Blog Posts". |
| 4 | Refine the visual design of the blog post pages by introducing a cohesive color scheme and typography style that enhances readability and aesthetic appeal. Use a serif font for blog titles and a sans-serif font for body text to create a clear hierarchy and visual contrast. Apply a soft background color to the entire page to reduce eye strain and highlight the content areas. Ensure that interactive elements such as social media sharing buttons, comment section, and "Like" buttons have a distinct hover effect to visually indicate interactivity. | Refine the visual design of the blog post pages by implementing a cohesive color scheme and typography style to enhance readability and aesthetic appeal. Introduce a serif font for blog titles and a sans-serif font for body text to establish a clear hierarchy and visual contrast. Apply a soft background color to the entire page to reduce eye strain and highlight the content areas. Ensure that interactive elements, such as the social media sharing buttons within each blog post, the comment section, and the "Like" buttons adjacent to comments, have a distinct hover effect to visually indicate interactivity. Note that if the distinct hover effect is not currently visible, it must be implemented to enhance user experience. |
| 8 | Introduce a "Related Posts" section at the end of each blog post page, featuring three blog posts with similar themes to engage readers further. Ensure that each related post displays a title, a brief excerpt, and a placeholder image, with links to navigate to the full blog post. Allow users to access this section by scrolling to the bottom of a blog post page, just above the comment section. Incorporate a feature that dynamically selects related posts based on shared tags or categories with the current post. | To enhance user engagement on the blog post pages, introduce a "Related Posts" section directly above the existing comments section on each blog post page. This section should display three posts that share similar themes with the current post, determined by matching tags or categories. Each related post should include a title, a short excerpt, and a placeholder image, all of which should be clickable links that take users to the full blog post. This feature will help keep readers interested and encourage them to explore more content on the site. Ensure this section is seamlessly integrated so that users can easily find it by scrolling to the end of the blog post content. |
| 10 | Enhance the visual design of the email subscription feature across the homepage and each blog post page by introducing a stylish subscription form. Use a rounded input field with a placeholder text that invites users to "Enter your email...". Add a vibrant call-to-action button next to the input field, styled with a hover effect that changes its background color. Ensure the form is positioned prominently within the layout, and add a subtle animation that draws attention to the form upon page load. | Refine the email subscription feature on the homepage and each blog post page by enhancing the subscription form's design. Implement a rounded input field with the placeholder text "Enter your email..." and position a vibrant call-to-action button next to it. This button should have a hover effect that changes its background color to make it more engaging. Ensure the form is prominently placed within the layout to capture attention and add a subtle animation that highlights the form when the page loads. Since the current form is basic, these enhancements are not yet implemented and must be addressed. |

Table 11: Example outputs for simulated textual instructions. Content in the instructions that are distinct from original user intents are highlighted in gold.

Listing 1: **Code in the *first* step in the visual instruction simulation process.** The resulted image is in Figure 9. Our custom tools are highlighted in gold.

```
import matplotlib.pyplot as plt
from PIL import Image
import matplotlib.patches as patches

# Define the number of subplots
fig, axes = plt.subplots(1, 2, figsize=(12, 6))

# First Subplot: Blog Post Page
axes[0].imshow(Image.open("screenshot_blog-post.png"))
scroll_shape = patches.Rectangle((8, 160.9), 809, 21, facecolor='none', edgecolor='#FF0000')
axes[0].add_patch(scroll_shape)
```

```
1134   text_annotation(axes[0], "Scroll down to comment section", scroll_shape, color='#FF0000')
1135   axes[0].set_title("Blog Post Page")
1136
1137   # Second Subplot: Comment Section with Like Feature
       html_code_comments = """
1138   <html lang="en"><head>
           <meta charset="UTF-8">
1139       <meta name="viewport" content="width=device-width, initial-scale=1.0">
1140       <title>Comments</title>
           <style>
1141           body { font-family: Arial, sans-serif; }
1142           .comment-section { margin-top: 20px; }
               .comment { margin-top: 10px; }
1143       </style>
1144   </head>
       <body>
1145       <h1>Comments</h1>
1146
           <div class="comment-section">
1147           <textarea id="comment" placeholder="Add a comment..." rows="4"
1148   cols="50"></textarea>
               <button onclick="postComment()">Post Comment</button>
1149
               <div id="comments-list" class="comments-list">
1150               <div class="comment">
1151                   <p>Sample Comment</p>
1152                   <button onclick="likeComment(this)">Like</button> 0 Likes
                       <button onclick="replyToComment(this)">Reply</button>
1153               </div>
1154           </div>
           </div>
1155   </body></html>
       """
1156   layout_visualization(html_code_comments, axes[1])
1157   axes[1].set_title("Comment Section with Like Feature")
1158
       plt.tight_layout()
1159
```

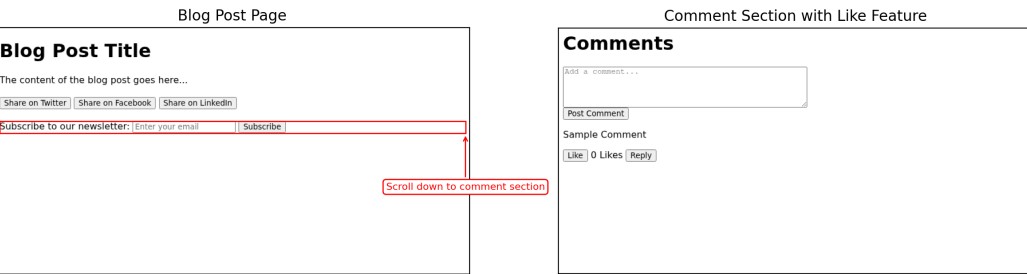

Figure 9: Intermediate drawing results after executing code in Listing 1.

Listing 2: **Code in the *second* step in the visual instruction simulation process.** The resulted image is in Figure 10. Our custom tools are highlighted in gold.

```
# Adding annotations for the "Like" feature in the comment section

# Coordinates for the "Like" button and "0 Likes" counter
like_button_coords = (8, 209.4, 42.6, 21)
like_counter_coords = (55.7, 209.4, 55.5, 19)

# Drawing a rectangle around the "Like" button
like_button_shape = patches.Rectangle(
    (like_button_coords[0], like_button_coords[1]),
    like_button_coords[2],
    like_button_coords[3],
    facecolor='none',
    edgecolor='#00FF00'
)
axes[1].add_patch(like_button_shape)
text_annotation(axes[1], "Add 'Like' button", like_button_shape, color='#00FF00')

# Drawing a rectangle around the "0 Likes" counter
like_counter_shape = patches.Rectangle(
    (like_counter_coords[0], like_counter_coords[1]),
```

```
        like_counter_coords[2],
        like_counter_coords[3],
        facecolor='none',
        edgecolor='#00FF00'
)
axes[1].add_patch(like_counter_shape)
text_annotation(axes[1], "Show like count", like_counter_shape, color='#00FF00')
```

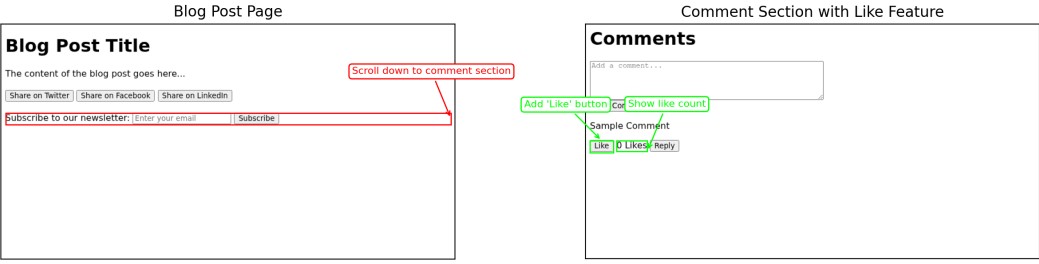

Figure 10: Simulated visual instructions for `t=2`. This is obtained after executing code in Listing 2 on top of Figure 9. User intent: Enhance the comment section by adding a "Like" feature to each comment and reply, allowing users to express appreciation for comments. Ensure that each comment and reply has a visible "Like" button next to it, which increments a visible like counter when clicked. Users should be able to navigate to the comment section by scrolling to the bottom of any blog post page from the homepage.

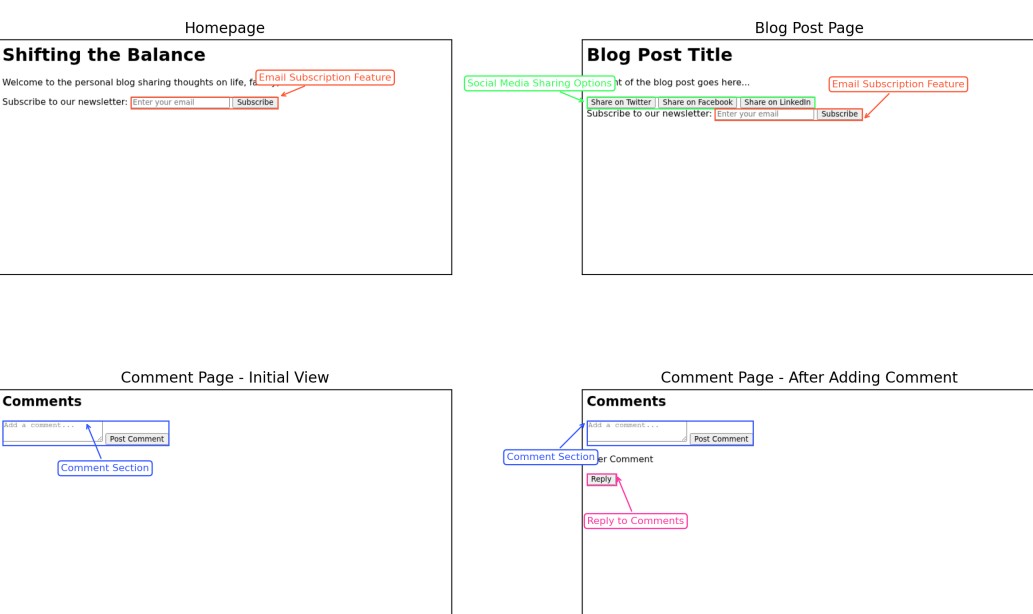

Figure 11: Simulated visual instructions for `t=1`. User intent: Build a website for "Shifting the Balance" that serves as a personal blog sharing thoughts on life, family, and societal issues. Implement a comment section accessible at the bottom of each blog post page, allowing users to leave comments and replies. Include an email subscription feature on the homepage and each blog post page, enabling visitors to subscribe with their email addresses to receive notifications about new posts. Provide social media sharing options for Twitter, Facebook, and LinkedIn on each blog post page to facilitate easy sharing of content.

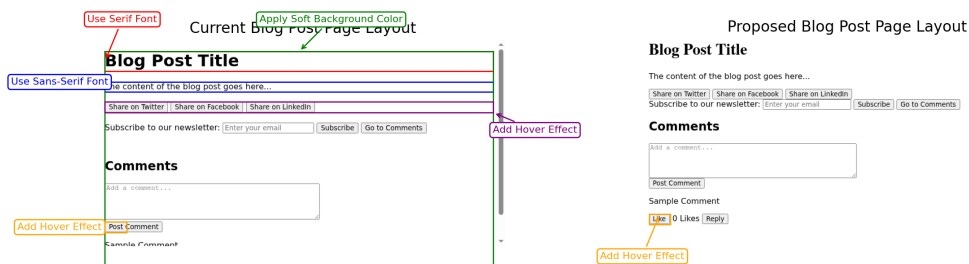

Figure 12: Simulated visual instructions for t=4. User intent: Refine the visual design of the blog post pages by introducing a cohesive color scheme and typography style that enhances readability and aesthetic appeal. Use a serif font for blog titles and a sans-serif font for body text to create a clear hierarchy and visual contrast. Apply a soft background color to the entire page to reduce eye strain and highlight the content areas. Ensure that interactive elements such as social media sharing buttons, comment section, and "Like" buttons have a distinct hover effect to visually indicate interactivity.

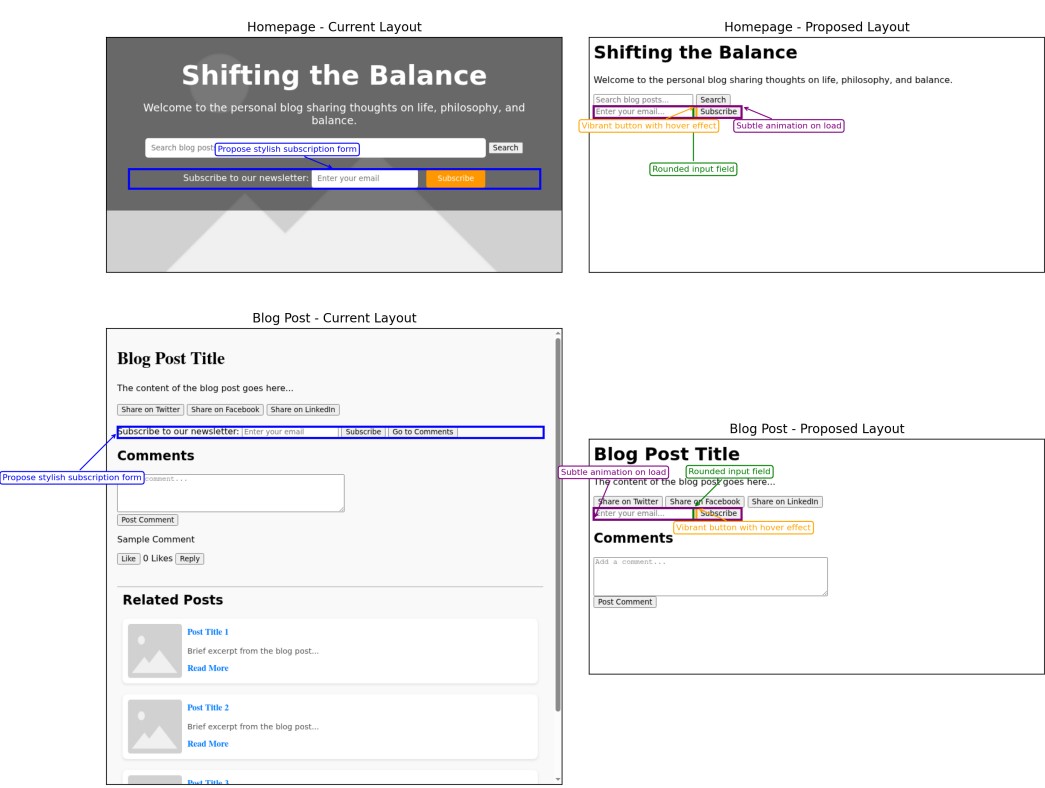

Figure 13: Simulated visual instructions for t=10. User intent: Enhance the visual design of the email subscription feature across the homepage and each blog post page by introducing a stylish subscription form. Use a rounded input field with a placeholder text that invites users to "Enter your email...". Add a vibrant call-to-action button next to the input field, styled with a hover effect that changes its background color. Ensure the form is positioned prominently within the layout, and add a subtle animation that draws attention to the form upon page load.

## D.3 AGENT-BASED EVALUATION

Following the WebVoyager framework (He et al., 2024), our evaluation agent interacts with a web environment through multiple steps. At each step, the agent receives two forms of input observations, as shown in Figure 14: (1) A screenshot of the web page, annotated with numerical labels on all interactive elements, and (2) A corresponding text representation listing each labeled element and its properties. Based on these observations, the agent selects an action from a predefined

action space: `Click`, `Hover`, `Type`, `Select`, `Scroll`, `GoBack`, and `Upload`. Most actions require a *target* argument, typically a numbered interactive element or the entire screen. Actions TYPE, SELECT, and `Upload` require an additional content argument, while GOBACK requires no arguments.

We additionally augment the agent with **image manipulation tools**. These tools do not interact directly with the web environment, but instead operate on screenshots from prior turns. They are designed to sharpen the agent's perception and assist in judging test cases that require fine-grained visual perception, such as verifying a subtle hover effect in a local region. The available tools are:

1. `ViewRaw screenshot_x`: view the high-fidelity screenshot of a specific step, without annotated set-of-marks that may potentially hinder evaluation;

2. `Compare [Numerical_Label]; screenshot_x, screenshot_y`: Compare a specific element across two different steps. This tool will crop the local regions for specified element [Numerical_Label] and concatenate the two crops across two screenshots into a single image for comparison.

3. `ViewAnimation [Numerical_Label or WINDOW]; screenshot_x`: View the sequence of screenshots during the transition to `screenshot_x` (from `screenshot_x-1`). Similar to Compare, this tool concatenate the images into a single large image for comparison.

Note that we use the *raw and high-resolution* screenshots in these tools, rather than the annotated and cluttered ones used for input observation. The full prompt for our augmented agent is in Listing 3. To show the benefits of our proposed image manipulation tools, Figure 15 shows an example where evaluator without image manipulation tools fails, but evaluator with image manipulation tools succeeds.

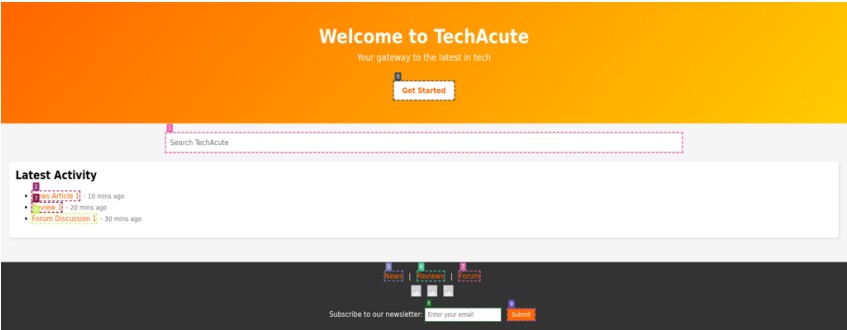

Figure 14: An example input observation for our evaluation agent. The textual representation is as follows: [0]: "Get Started"; [1]: <input> ""; [2]: "News Article 1"; [3]: "Review 1"; [4]: "Forum Discussion 1"; [5]: "News"; [6]: "Reviews"; [7]: "Forum"; [8]: ¡input¿ ""

## D.4 ACECODER

Detailed pseudo-code for ACECODER is presented in Alg. 2. In the algorithm, the feedback function AGENTFB($\mathcal{M}$, $\tilde{\mathbf{i}}$, $o$) $\rightarrow pass, expl$ queries a web agent powered by model $\mathcal{M}$ to verify whether instruction $\tilde{\mathbf{i}}$ is well implemented in output $o$, and provides a binary feedback $pass$ along with a text explanation $expl$.

An example from our test set demonstrating ACECODER is as follows. The simulated user instruction is – Enhance the "Local Attractions" page by adding a feature for users to create a personalized itinerary. This functionality will allow users to select attractions directly from the detailed view of each attraction and add them to their itinerary. Users should be able to include personal notes or preferences for each attraction they add. The itinerary should be accessible from the homepage, with options to view and print it. Ensure smooth navigation between the homepage and the hotel comparison page, allowing users to return without losing their itinerary details. Note that the itinerary feature does not currently exist, and thus must be implemented before these instructions can be fully applied.

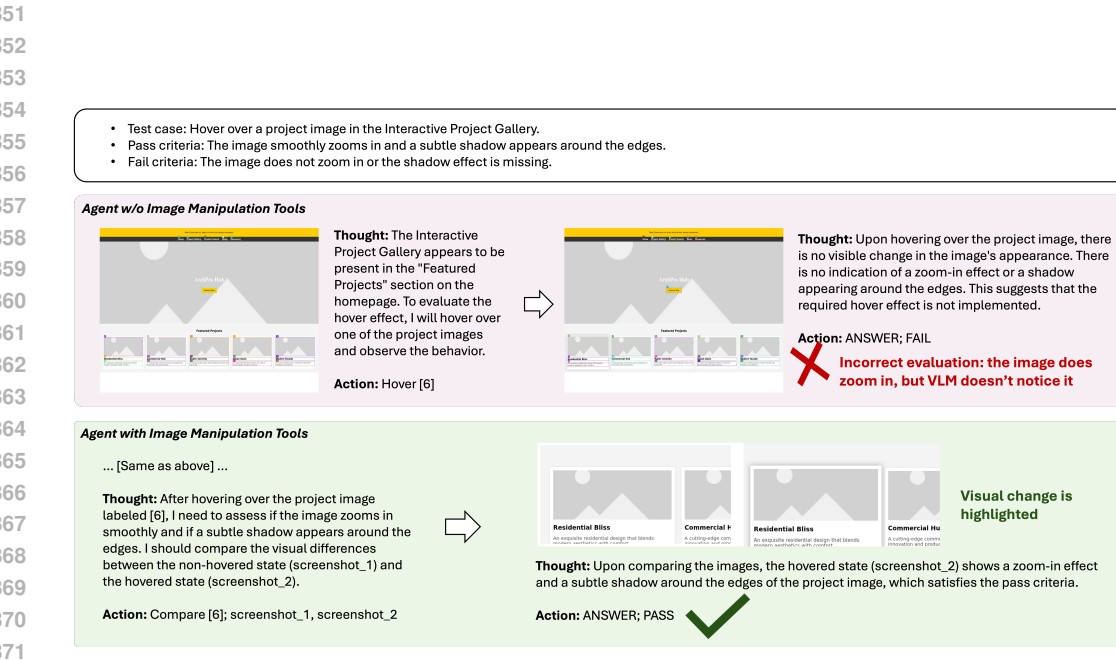

Figure 15: Example where evaluator without image manipulation tools fails, but evaluator with image manipulation tools succeeds.

---

**Algorithm 2 ACECODER algorithm at each turn.**

---

**Require:** Model $\mathcal{M}$; current-turn instruction $\tilde{\mathbf{i}}_t$; history $\mathbf{H}_t$
**Require:** Feedback function $\text{AGENTFB}(\mathcal{M}, \tilde{\mathbf{i}}, o) \rightarrow pass, expl$
**Ensure:** Website $w$
1: $o_t \leftarrow \mathcal{M}(H_{t-1}, x_t)$           ▷ Draft output for turn $t$
2: $F \leftarrow \emptyset$              ▷ Initialize feedback set
3: **for** $j \leftarrow 1$ **to** $t$ **do**
4:   $pass_j, expl_j \leftarrow \text{AGENTFB}\left(\mathcal{M}, \tilde{\mathbf{i}}_j, o_t\right)$
5:   **if** $pass_j = $ **false then**
6:    $F \leftarrow F \cup \{expl_j\}$         ▷ Collect textual feedback
7:   **end if**
8: **end for**
9: **if** $F \neq \emptyset$ **then**
10:   $o_t \leftarrow \mathcal{M}(\mathbf{H}_t, \tilde{\mathbf{i}}_t, F)$        ▷ Re-generate with feedback
11: **end if**
12: $w \leftarrow \text{RENDER}(o_t)$
13: **return** $w$

---

A common implementation pitfall for this instruction is that an "Add to Itinerary" button is added to the interface, but is not functional. For the first iteration of model output $\mathcal{M}(H_{t-1}, x_t)$ exhibiting this pitfall, the agent feedback $expl_t$ generated by ACECODER is as follows:

```
Summary:
1. Started on the homepage and navigated to the "Local Attractions" page.
2. Attempted to add attractions to a personalized itinerary using the "Add to Itinerary"
buttons, but these actions were not executable.
3. Accessed the "My Itinerary" page, which displayed an empty itinerary with only a "Print
Itinerary" button, and no options to add notes or preferences.
4. Returned to the homepage to verify if itinerary details persist across navigation, but
there was no indication of any preserved itinerary details.

Analysis:
- The website does not currently support the full functionality of creating a personalized
itinerary as instructed.
- The "Add to Itinerary" buttons do not function, and there is no way to add personal notes
or preferences.
- The itinerary is not populated with selected attractions, and there is no evidence of
itinerary persistence when navigating between pages.

Missing Elements:
- A functional "Add to Itinerary" feature that allows for adding attractions with personal
notes.
- Persistent itinerary details that remain accessible across different pages.
- A confirmation or feedback mechanism to indicate successful addition of attractions to the
itinerary.
```

With the user instruction enhanced by this agent feedback, the model successfully addresses this issue and generates functional "Add to Itinerary" buttons. Detailed screenshots illustrating the website before and after using the agent feedback is shown in Figure 16.

# E   ADDITIONAL QUANTITATIVE ANALYSIS

## E.1   AGREEMENT ANALYSIS ACROSS DIFFERENT LLMS AS EVALUATORS

In addition to examining the alignment between human evaluators and LLM judges in §3.3, we further assess the reliability of the evaluation pipeline by measuring the agreement across different LLMs as evaluators. We employ three LLMs including GPT-4o, Gemini-2.5-Pro and Qwen2.5-VL-72B, to evaluate a outputs from four representative models. For usability, we evaluate a subset of 355 test cases, while for usability we evaluate the entire test set. Detailed numbers are in Table 12. While the absolute scores varied slightly across judges, **the relative ranking of all models remains consistent across all three LLMs**, demonstrating robust agreement in model ranking and evaluation. We additionally calculated the IAA metrics for pass rate evaluation, achieving an accuracy of 81.4 (Cohen's $\kappa$ of 62.3), indicating **substantial agreement**. These results further demonstrate the reliability of our LLM-based evaluation framework.

|  |  | GPT-4o | Gemini-2.5-Pro | Qwen2.5-VL-72B | Qwen2.5-VL-7B |
|---|---|---|---|---|---|
| *GPT-4o Evaluator* | | | | | |
| | PR↑ | 52.7 **(rank=2)** | 70.1 **(rank=1)** | 45.1 **(rank=3)** | 19.7 **(rank=4)** |
| | UX↑ | 55.0 **(rank=2)** | 71.0 **(rank=1)** | 46.5 **(rank=3)** | 26.8 **(rank=4)** |
| *Gemini-2.5-Pro* | | | | | |
| | PR↑ | 42.8 **(rank=2)** | 61.7 **(rank=1)** | 34.5 **(rank=3)** | 16.6 **(rank=4)** |
| | UX↑ | 45.5 **(rank=2)** | 75.0 **(rank=1)** | 35.3 **(rank=3)** | 13.3 **(rank=4)** |
| *Qwen2.5-VL-72B Evaluator* | | | | | |
| | PR↑ | 46.5 **(rank=2)** | 56.1 **(rank=1)** | 38.0 **(rank=3)** | 20.6 **(rank=4)** |
| | UX↑ | 61.0 **(rank=2)** | 61.3 **(rank=1)** | 49.5 **(rank=3)** | 38.5 **(rank=4)** |

Table 12: We report evaluation results obtained by three different LLM evaluators, GPT-4o, Gemini-2.5-Pro, and Qwen2.5-VL-72B. We report the pass rate (PR) and usability (UX) for four representative set of model outputs, GPT-4o, Gemini-2.5-Pro, Qwen2.5-VL-72B, and Qwen2.5-VL-7B.

### E.2 CONFIDENCE INTERVALS OF EVALUATION

To assess the statistical reliability of our results, we conduct multiple evaluation runs for key experiments: GPT-4o and Qwen2.5-VL-72B, each tested both with and without AceCoder. The evaluation is performed on 382 test cases; Each configuration has been evaluated five times. The mean performance and corresponding 95% confidence intervals are reported in Table 13. The narrow confidence intervals indicate that the measurements are stable. Moreover, the confidence intervals for the ACE-CODER-enhanced models do not overlap with those of their respective baselines, demonstrating that the improvements introduced by ACECODER are statistically reliable despite evaluation variance.

|  | Pass Rate |
|---|---|
| GPT-4o | $65.1 \pm 3.7$ |
| +ACECODER | $71.2 \pm 2.9$ |
| Qwen2.5-VL-72B | $58.8 \pm 1.4$ |
| +ACECODER | $62.3 \pm 1.2$ |

Table 13: Confidence interval calculated by repeating evaluation for five times.

## F ADDITIONAL QUALITATIVE ANALYSIS

### F.1 ERROR CASES

An example for forgetting issue is shown in Figure 17. Additionally, we show three error cases for the three types of visual instruction interpretation errors in Figure 18, 19 and 20 respectively.

### F.2 EXAMPLE TRAJECTORY FROM USABILITY EVALUATION

Table 14: Example trajectory from usability evaluation.

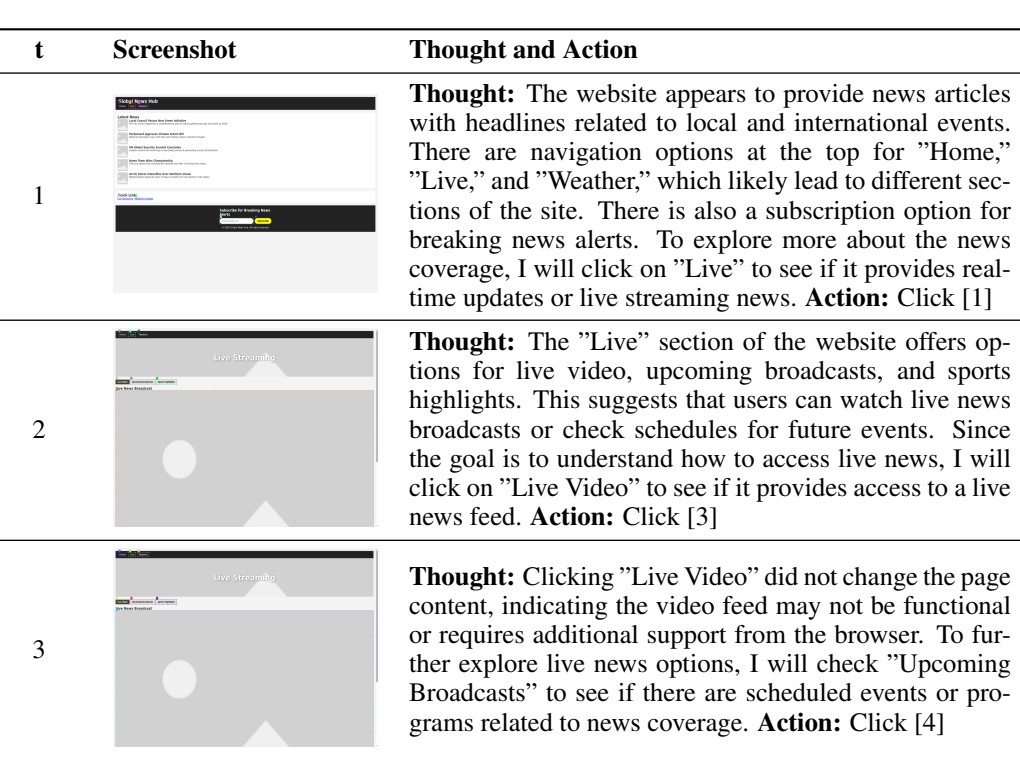

| t | Screenshot | Thought and Action |
|---|---|---|
| 1 | | **Thought:** The website appears to provide news articles with headlines related to local and international events. There are navigation options at the top for "Home," "Live," and "Weather," which likely lead to different sections of the site. There is also a subscription option for breaking news alerts. To explore more about the news coverage, I will click on "Live" to see if it provides real-time updates or live streaming news. **Action:** Click [1] |
| 2 | | **Thought:** The "Live" section of the website offers options for live video, upcoming broadcasts, and sports highlights. This suggests that users can watch live news broadcasts or check schedules for future events. Since the goal is to understand how to access live news, I will click on "Live Video" to see if it provides access to a live news feed. **Action:** Click [3] |
| 3 | | **Thought:** Clicking "Live Video" did not change the page content, indicating the video feed may not be functional or requires additional support from the browser. To further explore live news options, I will check "Upcoming Broadcasts" to see if there are scheduled events or programs related to news coverage. **Action:** Click [4] |

Table 14 – *Continued from previous page*

| t | Screenshot | Thought and Action |
|---|---|---|
| 4 |  | **Thought:** The "Upcoming Broadcasts" section provides a schedule of events, including news and weather updates, sports highlights, press conferences, global reports, and debates. Users can set reminders for these broadcasts, which is useful for keeping track of important events. To test the reminder functionality, I will set a reminder for the "Morning News & Weather" broadcast. **Action:** Click [8] |
| 5 |  | **Thought:** The pop-up message indicates that a reminder was successfully set for the "Morning News & Weather" broadcast. This feature enhances user engagement by allowing them to keep track of specific events. To further explore the website's offerings, I will check the "Sports Highlights" section to see if it provides specific content related to sports news. **Action:** Click [5] |
| 6 |  | **Thought:** The "Sports Highlights" section offers a variety of sports event recaps, including the World Cup Final, NBA Playoffs, UEFA Champions League, Formula One, and Olympic Athletics. Each event has a "Play" button, indicating video highlights are available. To test the functionality, I will click "Play" for the "2025 World Cup Final - Highlights" to view the content. **Action:** Click [6] |
| 7 |  | **Thought:** Clicking "Play" for the "2025 World Cup Final - Highlights" successfully opened a video player with a placeholder video. This confirms the website's functionality for viewing sports highlights. To return to exploring other sections of the site, I will close the video player. **Action:** Click [0] |
| 8 |  | **Thought:** The video player has been successfully closed, returning me to the "Sports Highlights" section. To explore other areas of news coverage, I will navigate to the "Weather" section to check if it provides current weather updates or forecasts. **Action:** Click [2] |
| 9 |  | **Thought:** The "Weather" section provides a severe thunderstorm warning, a 5-day forecast, current alerts, and an interactive radar. Users can also search for weather updates by location. To explore the current weather conditions, I will dismiss the severe weather warning message to have a clearer view of the page. **Action:** Click [6] |
| 10 |  | **Thought:** The severe weather warning has been dismissed, providing a clearer view of the weather section. There's a search bar available to enter a location for specific weather updates. To test this feature, I will input a location into the search bar to see detailed weather information for that area. **Action:** Type [6]; "New York" |

Table 14 – *Continued from previous page*

| t | Screenshot | Thought and Action |
|---|---|---|
| 11 | 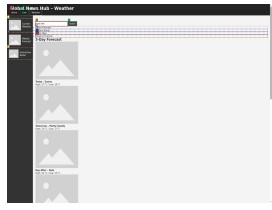 | **Thought:** I have entered "New York" into the search bar. To view the weather updates for New York, I need to click the "Search" button to apply the location filter and retrieve the relevant information. **Action:** Click [7] |
| 12 | 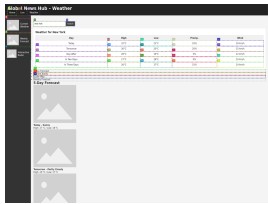 | **Thought:** After clicking the "Search" button, the weather section now displays detailed weather information for New York, including high and low temperatures, precipitation, and wind speed for the next five days. This confirms the site's ability to provide location-specific weather forecasts. To continue exploring the website's offerings, I will return to the "Home" section to see what general news coverage is available. **Action:** Click [0] |
| 13 | 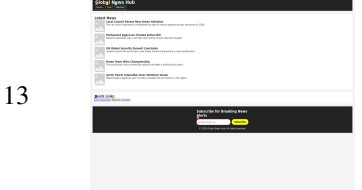 | **Thought:** Returning to the "Home" section, I see a list of the latest news articles covering various topics such as environmental initiatives, political actions, sports achievements, and weather events. There is also an option to subscribe for breaking news alerts, which is useful for users who want to stay updated. To test the subscription feature, I will enter an email address into the subscription textbox. **Action:** Type [5]; "user@example.com" |
| 14 | 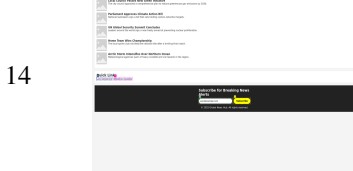 | **Thought:** I have entered the email address "user@example.com" into the subscription textbox. To complete the subscription process for breaking news alerts, I need to click the "Subscribe" button. **Action:** Click [6] |
| 15 | 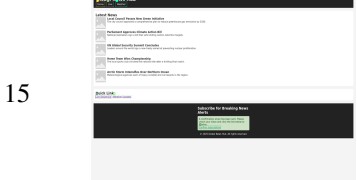 | Step limit reached |

At the end of the trajectory, we prompt the model to summarize the user experience throught the trajectory. The summary for the trajectory in Table 14 is as follows:

# G    PROMPTS

Listing 3: Prompt for evaluating **pass rate**.

```
You are an expert evaluator of built websites. You are given a test
condition along with its success/failure criteria. Your task is to
determine whether the website **PASS**es or **FAIL**s the test
condition. You can perform the evaluation either by examining a static
screenshot of the homepage or, if necessary, by navigating and
interacting with the website.

At each step, I will provide you with:
```

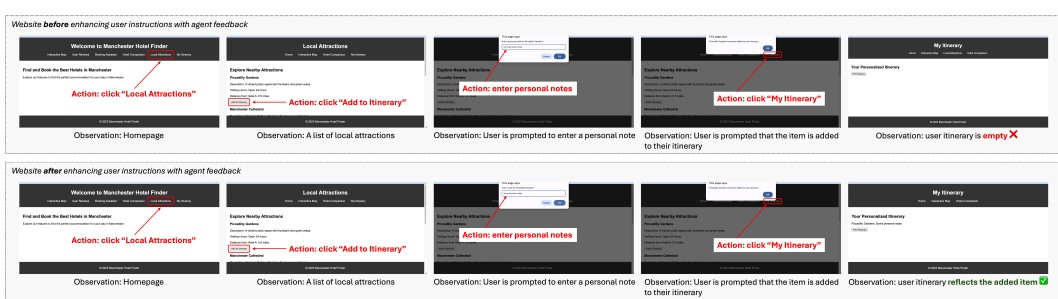

Figure 16: Screenshots of an example where ACECODER successfully addresses a pitfall in the implementation.

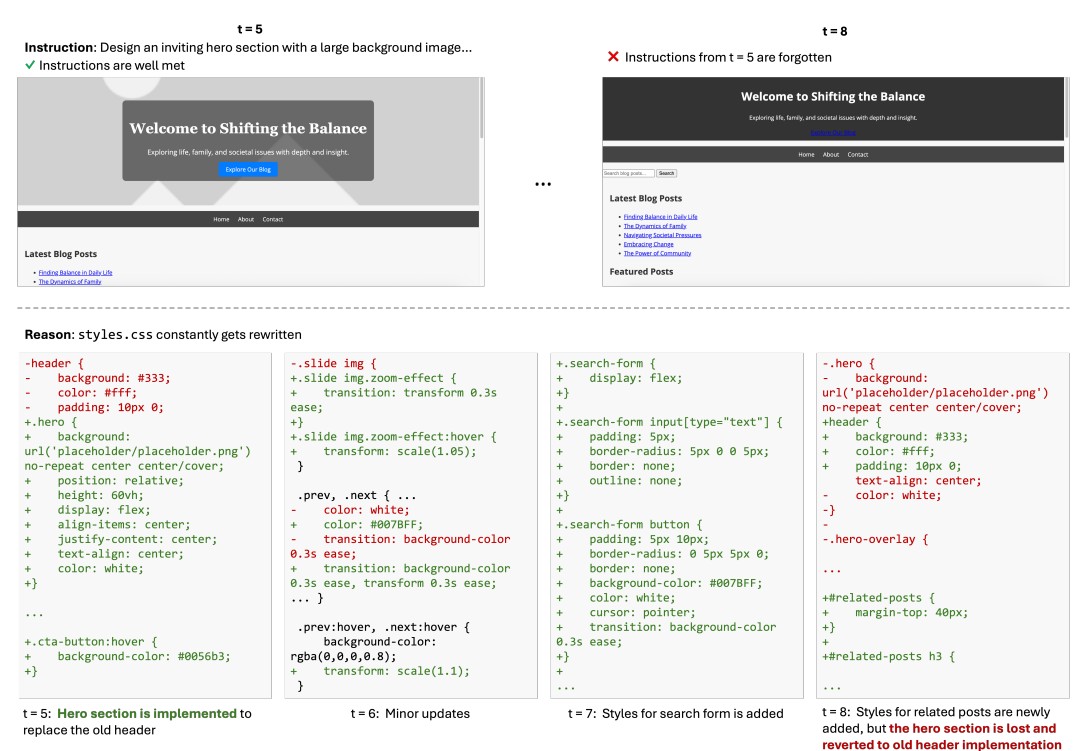

Figure 17: An example for forgetting issue. The hero section is correctly implemented in t=5, but gets forgotten at t=8. This is because the styles.css file constantly gets rewritten; at t=8, the rewriting lost the implementation of hero section, but reverts back to the old header implementation.

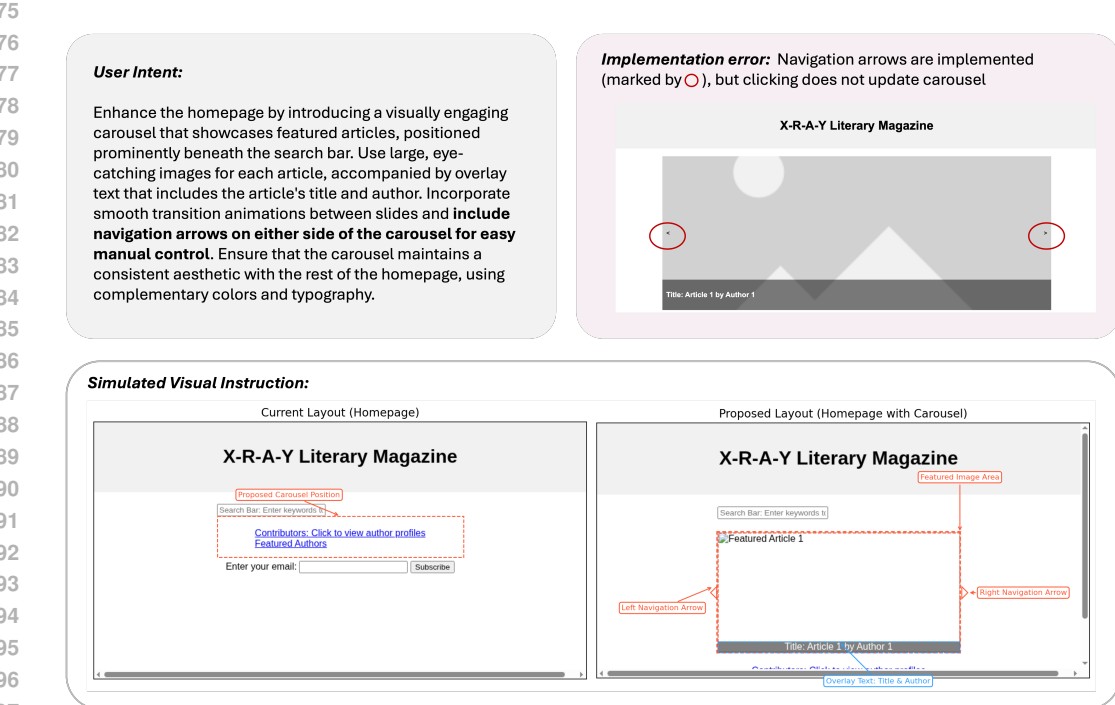

Figure 18: Error case for failure to implement implicit functionalities.

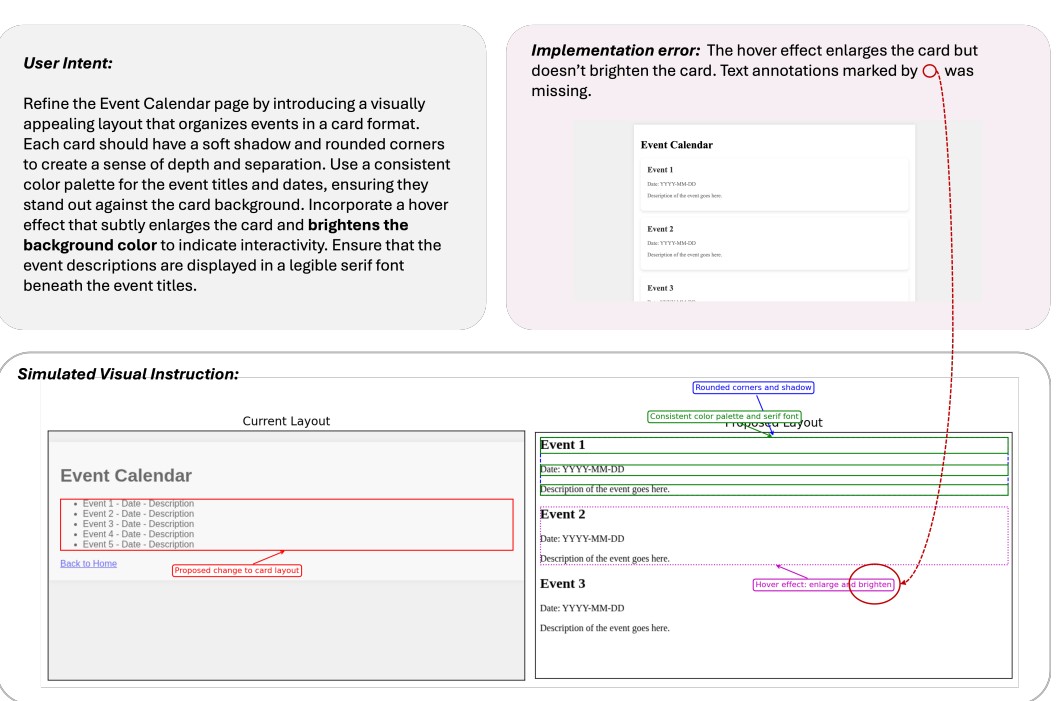

Figure 19: Error case for missing text annotations.

Figure 20: Error case for misinterpretation of visual clues.

```
The website provides a comprehensive platform for accessing news content across various
domains, including local and international news, weather updates, and sports highlights.
Here's a summary of my experience with the website:

1. **Navigation**: The main navigation options ("Home," "Live," "Weather") are clear and
intuitive, allowing users to easily switch between different sections of the website.

2. **Live Section**: The live section offers video content and schedules for upcoming
broadcasts, enhancing user engagement with live updates and reminders.

3. **Sports Highlights**: This section provides accessible video highlights for major
sporting events, adding value for sports enthusiasts.

4. **Weather Section**: The weather section includes current conditions, forecasts, and
alerts. It also features a location-based search function, which is practical for users
seeking specific weather information.

5. **Subscription Feature**: The option to subscribe for breaking news alerts is
straightforward, with clear feedback via a confirmation message upon successful subscription.

Overall, the website is user-friendly, with a clean layout and logical structure that
facilitates easy access to news content. The inclusion of multimedia content and interactive
features like setting reminders and searching for specific weather conditions further
enhances the user experience.
```

```
1. An annotated screenshot - with **numerical labels** placed in the
**top-left corner** of each web element. Each screenshot is named as
screenshot_x (x=1,2,...)
2. A simplified **textual representation** of the page -- including the
tag names and texts for every element annotated in the screenshot.

Your goal is to generate **ACTIONS** to perform the evaluation. If the
information is sufficient to make a judgment, your ACTION should be:
'ANSWER; PASS' or 'ANSWER; FAIL'. Otherwise, you should generate actions
to interact with the website before producing the answer. Choose one of
the following valid action formats:
```

```
### Valid Actions

Action should **STRICTLY** follow the format:
- Click [Numerical_Label]
- Hover [Numerical_Label]
- Type [Numerical_Label]; [Input_Text]
- Select [Numerical_Label]; [Option_Text]
- Scroll [Numerical_Label or WINDOW]; [up or down]
- GoBack
- Upload [Numerical_Label]; [Filename]
- ANSWER; [content]

Additional actions to inspect visual details, **STRICTLY** following the
format:
- Compare [Numerical_Label]; screenshot_x, screenshot_y
- ViewRaw screenshot_x
- ViewAnimation [Numerical_Label or WINDOW]; screenshot_x

### Action Guidelines

1. Execute only one action per iteration.
2. **Avoid repeating** the same action if the page does not change --
you may have chosen the wrong element or label.
3. To input text, you do **not** need to click the textbox first -- just
use the `Type` action. Pressing `ENTER` is handled automatically.
However, you may still need to click a search button afterward to apply
a filter.
4. Clearly distinguish between textboxes and buttons -- do **not** type
into a button. If no textbox is visible, consider clicking a search
button to reveal it.
5. To upload a file, you do **not** need to click the upload button
first -- just use the `Upload` action and specify the filename. The
filename **MUST** be chosen from: `placeholder.png`, `placeholder.mp4`,
`placeholder.mp3`, or `placeholder.pdf`.
6. Use `Compare` to compare the same element's visual display across two
**different** screenshots. For example, compare the screenshots before
(screenshot_x) and after (screenshot_y) a hover action to test hover
effect.
7. Use `ViewRaw` to retrieve the high-fidelity raw screenshot without
annotations.
8. Use `ViewAnimation` to view animated behavior when loading
screenshot_x. To focus on a specific element, provide its numerical
label.

### Evaluation Guidelines

1. You are also given the instructions for building the website, which
**MAY NOT** align with the actual website structure. Use them **only as
context** to help understand the website, but **always** rely on **real
interactions** with the website to make the final evaluation.
2. First verify whether the feature to be tested is present on the
website. If it is missing (e.g., the test condition refers to a header,
but the website does not have a header), the evaluation should be FAIL.
3. If the feature to be tested is not immediately visible on the
homepage but can be accessed from it, you must first navigate from the
homepage to find it. **Extensively** explore the website to locate the
feature, such as by scrolling down, checking menus, or following links,
until you confirm whether the feature exists.
4. After navigating to the feature, if the test condition is based on
**purely static** visual features such as color and layout, do not use
any more actions -- directly output `ANSWER` based on the screenshot.
5. When a test involves **multiple steps or questions**, use `ANSWER`
only **after** addressing all of them.
6. **Extensively** interact with the website to trigger behaviors
relevant to the test condition. For example, when testing a search bar,
```

```
test it with multiple inputs including both reasonable pseudo-data and
generic entries like "1", "2", "a", "b".
7. The annotated screenshot is **NOT** what end users actually see. To
evaluate the visual design, **always** rely on additional actions to
gather visual details before making the judgment.
8. The website uses placeholder data. Do **not** judge pass/fail
outcomes based on whether the displayed data reflects real-world data.
9. The website also uses placeholder content for media (images, videos,
audio, PDFs). Do **not** judge pass/fail outcomes based on the
**content** of these files. However, you should evaluate their display
and interactive behavior when determining pass/fail.

### Your Reply Format

Thought: {Step-by-step reasoning}
Action: {One properly formatted action}
```

Listing 4: Prompt for **simulating first-time users** in the evaluation of **usability**.

```
Your task is to evaluate the **usability** of a website. You will
simulate a **first-time user**: you are given only the website's
high-level goal without detailed instructions. Your task is to
extensively explore the website, infer what you want to accomplish as an
end user, attempt the tasks, and judge how easy it is to learn and use
the site to complete them.

At each step, I will provide you with:
1. An annotated screenshot - with **numerical labels** placed in the
**top-left corner** of each web element.
2. A simplified **textual representation** of the page -- including the
tag names and texts for every element annotated in the screenshot.

At each step, you can choose one of the following valid action formats:

### Valid Actions

Action should **STRICTLY** follow the format:
- Click [Numerical_Label]
- Hover [Numerical_Label]
- Type [Numerical_Label]; [Input_Text]
- Select [Numerical_Label]; [Option_Text]
- Scroll [Numerical_Label or WINDOW]; [up or down]
- GoBack
- Upload [Numerical_Label]; [Filename]

### Guidelines

1. Execute only one action per iteration.
2. **Avoid repeating** the same action if the page does not change --
you may have chosen the wrong element or label.
3. To input text, you do **not** need to click the textbox first -- just
use the `Type` action. Pressing `ENTER` is handled automatically.
However, you may still need to click a search button afterward to apply
a filter.
4. Clearly distinguish between textboxes and buttons -- do **not** type
into a button. If no textbox is visible, consider clicking a search
button to reveal it.
5. To upload a file, you do **not** need to click the upload button
first -- just use the `Upload` action and specify the filename. The
filename **MUST** be chosen from: `placeholder.png`, `placeholder.mp4`,
`placeholder.mp3`, or `placeholder.pdf`.
6. The website uses placeholder for data and media (images, videos,
audio, PDFs).

### Your Reply Format
```

```
Thought: {Describe image content, then perform step-by-step reasoning}
Action: {One properly formatted action}
```

Listing 5: Prompt for performing **pair-wise comparison** in the evaluation of **usability**.

```
You are an expert UX Researcher specializing in usability analysis. Your
analysis must be objective, evidence-based, and grounded in established
usability principles.

You will be given two user interaction trajectories, 'Trajectory A' and
'Trajectory B'. Your task is to perform a pairwise comparison to
determine which website offers a better user experience. You will
provide a detailed analysis and a final, single-word verdict for Website
A: WIN, LOSE, or TIE.

### Evaluation Criteria

Base your analysis *strictly* on the following usability dimensions:

1. **Journey Success:** Did a goal emerge from the user's browsing? If
so, how well did the website support their journey from discovery to
potential action? Did they successfully navigate to a satisfying
endpoint or did they abandon their exploration?
2. **Efficiency & Path:** How much effort (cognitive load, number of
actions, backtracking) was required? Was the user's path logical and
fluid, or was it disjointed and filled with unnecessary steps?
3. **Satisfaction & Friction:** How did the user feel? Identify specific
moments of frustration, confusion, delight, or ease, as indicated by
their thoughts.
4. **Discoverability & Clarity:** How easy was it for the user to
understand the site's structure, find what's available, and learn what
actions are possible? Is the value proposition clear from the start?
This is especially critical for **Exploratory** tasks.

### Instructions

Follow these steps and structure your response exactly as follows.

1. **Identify Emergent Goal(s) (if any):** Briefly state the primary
goal(s) the user developed during each trajectory.
2. **Analyze Trajectory A:** Summarize the user's experience on Website
A, citing specific evidence for each of the four evaluation criteria.
3. **Analyze Trajectory B:** Do the same for Website B.
4. **Comparative Analysis:** Directly compare Website A and Website B on
each of the criteria. State which website performed better and why.
5. **Final Verdict & Justification:** Conclude with a verdict for
Website A and a concise justification highlighting the most critical
factors. Formulate your final verdict as follows:

VERDICT: [WIN, LOSE, or TIE]
```

Listing 6: Prompt for **textual instruction simulator** for functionality-type user intents. The gold text refers to user intent, and the blue text refers to the current code state.

```
You are provided with instructions to **add or refine functionalities**
of a website. Your task is to refine the instructions so they are
accurate, specific, and actionable. Use the code of the existing website
as context.
1. **Verify component or functionality references.** These references
typically involve **where** to implement the new feature, or **how** to
navigate to the new feature from the homepage.
    * If a referenced component/functionality exists, refine vague
references to be more precise.
```

```
     * Example: Change "add a button to the main menu" to "add a button
to the dark gray navigation bar at the top".
    * If it does **not** exist, clearly note that it must be implemented
before the instruction can be applied.
2. **Clarify vague instructions.** If an instruction is not specific
enough, make it more concrete based on the actual website.
3. **Avoid code-level details.** Do not reference class names, HTML
attributes, color hex codes, or other implementation-specific
identifiers.

Your refined instructions should be a short paragraph (3-6 sentences).
Start your refined instructions with
**Response:**

# Instructions to Refine

USER INTENT

# Code for Existing Website

## filename1

file content 1

## ...
```

Listing 7: Prompt for **textual instruction simulator** for design-type user intents. The gold text refers to user intent, and the blue text refers to the current code state.

```
You are provided with instructions to **refine the visual design** of a
website. Your task is to refine the instructions so they are accurate,
specific, and actionable. Use the code of the existing website as
context.
1. **Check for already implemented instructions.** Compare each
instruction to the current state of the website. If an instruction is
already implemented, remove it from the refined list.
    * Example: If the instruction says "Use a two-column layout" but the
site already has a two-column layout, remove that instruction.
    * Keep all unfulfilled instructions exactly as they are - **do not
omit or change them** unless covered by steps 2-3 below.
2. **Verify component or functionality references.**
    * If a referenced component/functionality exists, refine vague
references to be more precise.
      * Example: Change "the button" to "the green submit button in the
bottom left".
    * If it does **not** exist, clearly note that it must be implemented
before the instruction can be applied.
3. **Clarify vague instructions.** If an instruction is not specific
enough, make it more concrete based on the actual website.
4. **Avoid code-level details.** Do not reference class names, HTML
attributes, color hex codes, or other implementation-specific
identifiers.

Your refined instructions should be a short paragraph (~5 sentences).
Start your refined instructions with
**Response:**

# Instructions to Refine

USER INTENT

# Code for Existing Website

## filename1
```

```
file content 1

## ...
```

Listing 8: Prompt for **visual instruction simulator**. The gold text refers to user intent. The screenshots for every rendered page along with their filenames (in the format of `screenshot_name.png`) are appended in the following messages.

```
You are an expert drawing agent with a series of drawing tools based on
`matplotlib`. Your task is to visualize a series of user instructions
about implementing or refining the {{functionality|visual design}} of a
website.

For each page in the website, you are given:
* A screenshot (with its filename),
* The HTML code, and
* The coordinates of elements in the page.

You can load and display a screenshot with:
```python
from PIL import Image
ax.imshow(Image.open("screenshot_name.png"))
```

# User Instructions

USER INTENT

# Tool Documentation

You have the following four tools:
1. Subplot organization
2. Layout visualization
3. Shape drawing
4. Text annotation

You'll generate python code to call these tools - I'll explain the tools
one by one. Your code should be wrapped within ``python ```. Make sure
to import necessary tools and libraries you want to use.

## Subplot Organization

When visualizing the instructions, you may need to work with multiple
subplots:
1. **First subplot - always the screenshot of relevant page**. Use
shapes and text annotations to link each instruction to its
corresponding component. For example:
   * If the instruction states, "Use a dark background color with white
text for the menu items", add annotations to the menu bar.
   * If it states, "Display the discount price in a bright color",
annotate the product card.
   * If it states, "Use a circular profile photo with a thin border",
annotate the profile picture accordingly.
2. **Revised layout subplot(s)** - for layout changes, if necessary.
   1. In the first subplot, highlight the components to be moved and
indicate their intended movement.
   2. In the later subplot(s), show the components in their **new
positions** with shapes and text annotations - using the **same
annotation color** for each component in both the original screenshot
and the updated layout views for clarity.
   3. To show the updated layout: (1) If the UI changes are minor, reuse
the screenshot and mark it with shapes showing the new positions. (2)
```

For major changes, use the HTML layout visualization tool to create a simplified updated layout.
3. **State transition subplot(s)** – for interaction-driven changes, if necessary
    1. In the first subplot, highlight the interactive component(s) that trigger the transition and annotate interaction type.
    2. In the new subplot, show the resulting end state and clearly indicate what changed.
4. **Titles**: Each subplot should have a **clear and descriptive title** to clearly indicate the relationship and transitioning between subplots (e.g., "Current Layout", "Proposed Layout", "After Clicking 'Submit'").
5. **Multiple pages**: If the instructions involve multiple pages, you can allocate one or multiple subplots for each page following the instructions above. Annotations for each page will be based on its own screenshot.

You may create subplots **only in your first drawing turn**. Before starting, decide exactly how many subplots you need and what **specific page state** each will represent. Once created, each subplot is fixed to a single, distinct view – you cannot add or remove subplots later.

Use matplotlib to create multiple subplots, and use each element of `axes` for further drawing:
```python
fig, axes = plt.subplots(n_row, n_col)
```

Note: For each subplot, **NEVER** directly draw on top of blank canvas background! You **must** define a background using one of the following methods:
1. Screenshot background: If you are visualizing the existing page or interaction flow, use the screenshot:
`ax.imshow(Image.open("current_screenshot.png"))`
2. HTML layout background: If you are visualizing a new layout or interface, use the layout visualization tool:
`layout_visualization(html_code, ax)`. After this call, wait for coordinate data before adding shapes or annotations.

## Layout Visualization

To visualize the components required in the instructions, you should create a **minimal** HTML layout **only use boxes and text**. Avoid colors, fonts, and detailed styling.

**Important**: You may receive reference HTML in the input. These are for context only – **do not** copy or reuse them in your `layout_visualization` call.

Use the following to visualize HTML layout in a given `ax`:
```python
html_code = """<HTML code here>"""
layout_visualization(html_code, ax)
```
Assume the function `layout_visualization` is already implemented – do not redefine it. After you call it, I will provide coordinates for each element in the next round. You can then use those to draw shapes or text annotations in future steps. **Do not** perform any annotations on `ax` before receiving the coordinates.

## Shape Drawing

Use `matplotlib.patches` to draw rectangles, circles, ellipses, or polygons. Shapes are typically used for: (1) Highlighting specific regions for which you have instructions; or (2) Indicating the addition

```
of new components, such as a circle for an icon or a rectangle for a new
block.

Examples:
```python
new_shape = patches.Rectangle((x, y), width, height, facecolor='none',
edgecolor='#??????')  # draw rectangle
new_shape = patches.Circle((x_center, y_center), radius,
facecolor='none', edgecolor='#??????')  # draw circle
new_shape = patches.Ellipse((x_center, y_center), width, height,
facecolor='none', edgecolor='#??????')  # draw ellipse
new_shape = patches.Polygon([[x1, y1], [x2, y2], [x3, y3]], closed=True,
facecolor='none', edgecolor='#??????')  # draw polygon
```

and then draw the shape:
```python
ax.add_patch(new_shape)
```

Shape drawing is usually paired with text annotation – you first draw a
shape to highlight a specific component, then annotate it with text.
When doing so, follow these color guidelines for clarity and visual
coherence:
1. **Consistent color for pairing**: Use the same `edgecolor` for the
shape and `color` for the text annotation to visually link them as a pair
2. **Shared annotation across components**: If a single annotation
(e.g., "add shadow effect") applies to multiple components, avoid
repeating the same annotation for each one. Instead, draw shapes around
**all relevant components** using the same distinctive color, and
annotate **just one** of them with a note explaining that the annotation
applies to all shapes of that color.
3. **Distinct colors for distinct meanings**: If you are applying
**multiple different annotations**, use **different colors** for each
group of shape/text pairs to clearly distinguish between them.

## Text Annotation

To annotate text, **NEVER use `ax.text` or `ax.annotate` directly**!
**Always** use the following function:
```python
text_annotation(ax, text, new_shape, color='#??????')
```
* `new_shape` should be a Rectangle, Circle, Ellipse, or Polygon (as
described in the Shape Drawing section) you want to annotate.
* This function draws an arrow pointing to the shape, with a labeled box
at the arrow's origin. The box and arrow will both use the specified
`color`, creating a visually consistent annotation.
* Placement and overlap are automatically managed to ensure clarity and
avoid visual clutter.
* The `text_annotation` function is already implemented – do not
redefine it.

## Multi-Turn Tool Calling

You can complete the drawing in a single turn or across multiple turns.
After each turn, I will provide the current figure. If you've called
`layout_visualization` in the turn, I will also provide the coordinates
for every element in the HTML – so if necessary, you can draw
annotations over the HTML visualization.

Most tasks can be completed in one turn. However, if you need to
annotate elements based on coordinates of HTML layout, it is acceptable
to use multiple turns.
```

