# OpenReview forum: "FronTalk: Benchmarking Front-End Development as Conversational Code Generation with Multi-Modal Feedback"
_ICLR.cc/2026/Conference — Submitted to ICLR 2026_

### Official Review · Reviewer_VZoF · 2025-10-26

**Soundness:** 3
**Presentation:** 4
**Contribution:** 3
**Rating:** 4
**Confidence:** 5

**Summary:**

This paper presents a benchmark that evaluates multi-turn and multi-modal code generation for front-end code generation. As  multi-modal feedback (textual, visual, and annotated screenshots) is critical for this target domain, the proposed benchmark is unique and useful.

**Strengths:**

* Multi-Modality: Assess textual and visual feedback, crucial for front-end development.
* Interesting analysis: Reporting analysis on implementation accuracy and user experience
* Baseline model: Discussing benchmarks with baseline model helps appreciating the valuae of benchmarks. e.g.,  "Forgetting Issue" in multi-turn generation.

**Weaknesses:**

*Time/infra cost: Sharing time/infra cost for benchmarks will be useful
*Public access: Any public link to reach benchmarks and report model performance would be useful

**Questions:**

Please answer questions in Weakness

---

> ### Author Response · Authors · 2025-12-03
>
> > ### [W1] Sharing time/infra cost for benchmarks will be useful
>
> Thank you for the suggestions! We have measured the cost for each major component of our pipeline using GPT-4o on the full dataset. The breakdown is as follows:
>
> | | # LLM requests | # Input tokens | # Cached input tokens | # Output tokens | Est. cost |
> |---|---|---|---|---|---|
> | Inference (textual) | 1.0k | 147k | 7.2M | 2.0M | $29.43 |
> | Inference (visual) | 1.0k | 1.2M | 9.9M | 1.3M | $27.95 |
> | User simulation (textual) | 900 | 3.6M | 0 | 115k | $10.11 |
> | User simulation (visual) | 2.0k | 16.1M | 12.7M | 1.0M | $66.38 |
> | Evaluation of pass rate | 13.3k | 16.2M | 36.4M | 893k | $95.00 |
> | Evaluation of usability | 1.6k | 2.2M | 3.8M | 258k | $12.75 |
>
>
> The evaluation components based on web agents account for the largest share of cost, followed by the visual user simulation that relies on a tool-use drawing agent. **This largely reflects the broader efficiency limitations of current LLM agents**, which involve multi-turn interactions with the environment, high-complexity observation spaces, and long multi-modal histories. Potential future directions for improving efficiency include more efficient agent designs, better history-compression mechanisms, and improved planning algorithms. We have added the details and discussion to Appendix C.2.
>
> > ### [W2] Public access: Any public link to reach benchmarks and report model performance would be useful
>
> Thanks for the important point! We have updated the supplementary materials for code and data. We will publicly release the dataset, codebase, and a full leaderboard upon publication.

---

### Official Review · Reviewer_SnEG · 2025-10-30

**Soundness:** 2
**Presentation:** 3
**Contribution:** 3
**Rating:** 4
**Confidence:** 3

**Summary:**

This paper introduces FronTalk, a benchmark for conversational code generation with multi-modal feedback, targeting front-end development (i.e., website generation).
The dataset is derived from the C4 corpus, containing: 100 multi-turn dialogues with 1K turns and 3.6K manually refined test cases.
An agent-based evaluation framework is proposed, where a web agent simulates users to explore generated websites and measures the functional correctness and the user experience (by pairwise way).
The authors evaluate 14 models and identify two key challenges: (i) a forgetting issue and (ii) a difficulty in interpreting visual feedback.
As a baseline, the authors propose AceCoder, an autonomous web agent that critiques the implementation of every past instruction, reduces the forgetting rate to nearly zero and improves performance by up to 9.3\% in pass rate.

**Strengths:**

- The paper presents a new benchmark focusing on multi-turn, multi-modal front-end development, accompanied by broad model evaluations and several meaningful analyses.

- The proposed AceCoder baseline effectively mitigates the forgetting issue.

**Weaknesses:**

- The main distinction from WebGen-Bench lies in the multi-turn setup. However, recent works such as WebGen-Agent [1] have already extended single-turn benchmarks into multi-turn scenarios, which suggests that this contribution may not be as technically challenging as implied.

- In Section 3.2, while Cohen's kappa is informative for evaluator reliability, it would be more meaningful to analyze intra-model (across different configurations) and inter-model (across models under the same configuration) ranking consistency. This aspect is missing but critical for validating benchmark stability.

- The AceCoder mechanism seems conceptually similar to running a single-turn setting in the main table--a concatenation of all user intents, effectively simplifying multi-turn reasoning into a static aggregation problem.
    - Notably, only 5 of the 14 evaluated settings (e.g., GPT-4o, Qwen2.5-VL-72B, Qwen2.5-VL-7B) show multi-turn (T) outperforming single-turn in pass rate.
    - For models where single-turn performance exceeds multi-turn (e.g., Claude-4-Sonnet, Llama-3.3-70B, GLM-4.1V-Thinking-9B, Ovis2.5-9B), it remains unclear whether AceCoder still provides statistically meaningful improvements.
    - If not, reverting to a single-turn setup might actually be the optimal strategy—rendering AceCoder unnecessary.

[1] Lu, Zimu, et al. "WebGen-Agent: Enhancing Interactive Website Generation with Multi-Level Feedback and Step-Level Reinforcement Learning." arXiv preprint arXiv:2509.22644 (2025).

**Questions:**

- The description of the single-turn baseline (concatenating all user feedback) only appears around line 154, which is quite late. This should be introduced earlier, ideally in Section 2.1. Additionally, including a variant that uses only the initial user intent (without concatenated feedback) would allow for an ablation study on the utility of simulated user feedback.

- In lines 459–464, the distinction between FronTalk and Sketch2Code should be elaborated more concretely. For example, show comparative examples illustrating how FronTalk involves more complex, realistic, and context-dependent interactions.

- Since both PR and UX automatic evaluations only achieve around 80% agreement with human decisions, consider reporting confidence intervals or error bars to indicate statistical significance when claiming improvements.

- Claude-4-Sonnet performs remarkably well in single-turn but poorly in multi-turn settings. Could this discrepancy relate to max token length (as suggested in Table 6, where Claude uses 30K tokens)? Re-evaluating it under shorter context windows (e.g., 10K–20K) could clarify this effect.

- Since 9 out of 14 models achieve higher pass rates in the single-turn setting, the results may reflect degradation from long iterative contexts. It would be valuable to discuss this phenomenon—similar findings were reported in recent studies [2]—as it might indicate a broader limitation of long-horizon LLM reasoning.

- (Not a requirement) I wonder if there are plans to release the training set (e.g., gym-like setup) to facilitate reproducibility and further research.


[2] https://alexzhang13.github.io/blog/2025/rlm/

---

> ### Author Response · Authors · 2025-12-03
>
> > ### [W1] Recent works such as WebGen-Agent have already extended single-turn benchmarks into multi-turn scenarios.
> Thank you for bringing up the concurrent work of WebGen-Agent. We have added discussions of this work in Section 5. While both efforts address front-end development and involve multi-turn interactions, **the core objectives and settings are fundamentally different**, which also underscores our unique contribution.
> 1. **FronTalk: a conversational benchmark centered on evolving user requests.** Our work introduces a conversational benchmark where **each turn introduces a new, distinct user request** (e.g., "build a blog" $\to$ in a later turn, "add a like feature to the comments"). This simulates a realistic collaboration where user requirements naturally evolve in response to the ongoing work. The core difficulty of FronTalk lies in understanding long contexts and satisfying sequential intents while maintaining previously established functionality. We provide a data example in Appendix B for clarity.
> 2. **WebGen-Agent: a self-refinement method towards a *fixed* initial request**. In contrast, WebGen-Agent is a **method** built on top of the single-turn WebGen-Bench. Its multi-turn process takes **the same initial instruction** and iteratively refines the solution based on LLM-generated feedback. These turns do not represent new user goals; instead, they form an automated feedback loop to refine outputs against the static initial requirements.
>
> Beyond its conversational design, FronTalk introduces some additional novelty:
> 1. **Visual feedback**: FronTalk features complex visual feedback grounded on prior dialogue context, capturing the dynamics of multi-turn interactions;
> 2. **Evaluation pipeline**: We adopt a novel evaluation paradigm based on web agent and introduce two complementary metrics: the pass-rate metric for instruction following and the usability metric capturing the frequently overlooked user-experience dimension.
> > ### [Q1] The description of the single-turn baseline (concatenating all user feedback) should be introduced earlier, ideally in Section 2.1.
> Thank you for the suggestion. We have added a clarifying paragraph in Section 2.1 to introduce the single-turn baseline.
>
> We wish to clarify its conceptual purpose, as **it is not intended as a directly comparable substitute for the multi-turn setting**, but an important analytical reference point. A practical multi-turn conversational scenario, where **new intents emerge in response to the evolving dialogue context**, cannot be faithfully reduced to a single concatenated prompt prepared in advance without considering that evolving context. The multi-turn setting captures the **dynamic, context-dependent nature** of real user collaboration, which is the core focus of our benchmark.
>
> The single-turn baseline instead serves two key **analytical purposes**:
> 1. **It benchmarks the inherent task difficulty:** By removing the complexity of multi-turn interaction, this setting measures the intrinsic challenge of the user intents and provides a baseline to which multi-turn performance can be compared.
> 2. **It contributes to a key research question:** Is it easier for models to process all requirements at once, or incrementally with multiple reasoning steps? Comparing the performance between the two settings yields mixed results, revealing distinct challenges for each setting. While the single-turn setting suffers from overly long and complex instructions, the multi-turn setting suffers from forgetting issues and long-context challenges. These findings highlight distinct failure modes, offering helpful insights for future investigation.
> > ### [Q1] A variant that uses only the initial user intent would allow for an ablation study on the utility of simulated user feedback.
> Regarding the suggested baseline, we wish to clarify that it would not serve as a valid baseline, as **later turns introduce new, distinct user intents**. Therefore, an LLM given only the first user intent would lack the crucial information necessary to fulfill subsequent requests, resulting in low performance on the full set of test cases.
>
> To illustrate this, we evaluated three representative models using only the initial intent. As the table shows, these variants achieve high pass rates on the subset of test cases corresponding only to the first turn, but fail on most test cases for later intents, since those later requirements were never communicated to the model. In contrast, the multi-turn variant performs consistently well on the complete test set, but its performance on first-turn test cases slightly degrades due to forgetting.
>
> | |Pass rate (first turn)|Pass rate (all turns)|
> |---|---|---|
> |GPT-4o (multi-turn)|58.8|**56.0**|
> |GPT-4o (first turn only)|**62.9** |16.4|
> |Gemini-2.5-Pro (multi-turn)|71.4|**75.0**|
> |Gemini-2.5-Pro (first turn only)|**77.2**|25.2|
> |Qwen2.5-VL-72B (multi-turn)|42.6|**52.8**|
> |Qwen2.5-VL-72B (first turn only)|**48.4**|12.1|

---

> ### Author Response · Authors · 2025-12-03
>
> > ### [W3] The AceCoder mechanism seems conceptually similar to running a concatenation of all user intents.
>
> We respectfully disagree that AceCoder is conceptually similar to a single-turn setting. AceCoder is based on two key designs, both fundamentally distinct from simple instruction concatenation:
> 1. **Agent-based critique**: AceCoder employs a web agent to interact with the rendered website, collect feedback, and accordingly regenerate code. This novel self-improvement mechanism is **orthogonal to the multi-turn formulation**. When used in a standalone manner only on the final turn, it still improves the performance from 56.0 to 61.0, as shown in our ablation study ("- Critique Past Turns" in Table 5).
> 2. **Per-turn critique**: AceCoder critiques the current website state independently against each prior instruction, summarizes the results, and feeds the critique summaries as input to the code-generation step. This design addresses the **forgetting issue** by providing the model with concise and targeted feedback grounded on the current website state. In contrast, a simple concatenation of all user instructions would rely on the model’s ability to reason correctly over a long, static context, and thus can often become bottlenecked.
>
> In our evaluation, we have compared AceCoder against two related baselines: (1) a **single-turn baseline** where all user intents are concatenated into a single input and processed in one turn, and (2) a **RePrompt** baseline, which also generates code in multiple turns, but replays a concatenation of all prior instructions in each turn. As shown in Section 4 and Table 5, AceCoder (pr=65.3, ux=76.3) significantly outperforms the single-turn baseline (pr=51.4, usability=50.0) and the RePrompt baseline (pr=62.3, ux=63.8), though RePrompt also brings sizable benefits. We have updated Table 5 to make the comparison clearer.
>
> > ### [W3] For models where single-turn performance exceeds multi-turn, it remains unclear whether AceCoder still provides statistically meaningful improvements.
>
> Thank you for this insightful question. We agree that identifying the conditions under which AceCoder provides meaningful benefit is essential. To investigate this, we evaluate AceCoder with Ovis2.5-9B whose multi-turn setting underperforms its single-turn counterpart. Results show that AceCoder *hurts* the performance of Ovis2.5-9B, yielding a pass rate of 18.1 compared to 34.5 in the baseline. Further analysis shows that Ovis2.5-9B struggles substantially during the agent-based critique step: it frequently forgets early instructions, fails to follow the format constraints specified in the system prompt, and quickly loses track of the critique objective.
>
> This highlights an important prerequisite: **AceCoder’s benefits depend on the base model possessing at least a baseline ability to reason and remain coherent across multi-turn interactions**. For models that already meet this bar (e.g., GPT-4o, Qwen2.5-VL-72B), AceCoder produces substantial improvements. As multi-turn reasoning capabilities of base models continue to advance — which we view as a likely trend — the value of agent-based critique mechanisms like AceCoder will continue to grow.
>
> > ### [W2] In Section 3.2 (analysis of automatic evaluation), it would be meaningful to analyze ranking consistency.
>
> Thank you for this valuable suggestion. We agree that analyzing ranking consistency is important for validating the benchmark's stability.
>
> Given the resource constraints of comprehensive human evaluation across all models, we instead assessed **cross-LLM ranking consistency** by employing three different LLM judges (GPT-4o, Gemini-2.5-Pro, and Qwen2.5-VL-72B) on a representative subset of models. The results show that, while the absolute scores slightly vary, **the relative ranking of all models remains consistent across all three LLMs**. Furthermore, the inter-annotator agreement between these LLM judges for pass rate reached an accuracy of 81.4 and Cohen's $\kappa$ of 62.3, indicating **substantial agreement**. Detailed setups are updated in Appendix E.1.
>
> | Evaluator$\to$ | GPT-4o | | Gemini-2.5-Pro | | Qwen2.5-VL-72B | |
> |---|---|---|---|---|---|---|
> | Models to be evaluated$\downarrow$ | **Pass rate** |**Usability** |**Pass rate** |**Usability** | **Pass rate** |**Usability** |
> | GPT-4o | 52.7 (rank=2) | 55.0 (rank=2) | 42.8 (rank=2) | 45.5 (rank=2) | 46.5 (rank=2) | 61.0 (rank=2) |
> | Gemini-2.5-Pro | 70.1 (rank=1) | 71.0 (rank=1) | 61.7 (rank=1) | 75.0 (rank=1) | 56.1 (rank=1) | 61.3 (rank=1) |
> | Qwen2.5-VL-7B | 19.7 (rank=4) | 26.8 (rank=4) | 16.6 (rank=4) | 13.3 (rank=4) | 20.6 (rank=4) | 38.5 (rank=4) |
> | Qwen2.5-VL-72B | 45.1 (rank=3) | 46.5 (rank=3) | 34.5 (rank=3) | 35.3 (rank=3) | 38.0 (rank=3) | 49.5 (rank=3) |

---

> ### Author Response · Authors · 2025-12-03
>
> > ### [Q2] The distinction between FronTalk and Sketch2Code should be elaborated more concretely.
>
> We have added a dedicated comparison with concrete examples in Figure 8. Compared to prior front-end benchmarks with image inputs e.g., Sketch2Code, Design2Code, FronTalk presents visual inputs that are realistic, complex and grounded on context, bringing unique challenges.
>
> > ### [Q3] Reporting confidence intervals or error bars to indicate statistical significance to ensure reliability of evaluation.
>
> To assess the statistical reliability of our results, we conduct multiple evaluation runs for key experiments: GPT-4o and Qwen2.5-VL-72B, each tested both with and without AceCoder. Each configuration has been evaluated five times. The mean performance and corresponding 95% confidence intervals are reported below:
>
>
> | Model                  | Pass Rate        |
> |------------------------|------------------|
> | GPT-4o                 | 65.1 $\pm$ 3.7       |
> | $~~$+ AceCoder          | 71.2 $\pm$ 2.9       |
> | Qwen2.5-VL-72B         | 58.8 $\pm$ 1.4       |
> | $~~$+ AceCoder          | 62.3 $\pm$ 1.2       |
>
>
>
> **The narrow confidence intervals indicate that the measurements are stable.** Moreover, the confidence intervals for the AceCoder-enhanced models do not overlap with those of their respective baselines, demonstrating that **the improvements introduced by AceCoder are statistically reliable despite evaluation variance**. This analysis has been added to Section E.2.
>
> > ### [Q4] Claude-4-Sonnet performs remarkably well in single-turn but poorly in multi-turn settings. [Q5] This discrepancy may relate to max token length and reflect degradation from long iterative contexts.
>
> We re-evaluated Claude-4-Sonnet under different token limits and observe a consistent pattern: **reducing the context window improves multi-turn performance**, even with an extremely low token limit of 10k that truncates 88% of model outputs. This supports the hypothesis that long-context degradation is a primary cause of Claude-4-Sonnet’s low performance in multi-turn interactions.
>
> | | Output truncation rate (%) | Pass rate | Usability |
> |---|---|---|---|
> |Multi-turn, token limit = 30k|5|42.0|52.1|
> |Multi-turn, token limit = 20k|23|44.6|50.0|
> |Multi-turn, token limit = 10k|88|47.8|54.2|
> |Single-turn|6|76.1|73.5|
>
> However, all multi-turn results still significantly underperform the single-turn baseline. A turn-by-turn analysis reveals the underlying issue: **performance is strong in early turns but declines sharply in later turns as the context grows longer**. This pattern further validates the hypothesis that context-length degradation is the primary cause.
>
> We thus identify **long-context processing as a key challenge for the multi-turn front-end development task,** as the multi-turn user-LLM interactions inevitably result in excessively long context. We observe similar degradation in other highly capable models, including DeepSeek-R1 and GLM-4.1V-Thinking, whereas models like GPT-4o and Gemini-2.5-Pro are not heavily impacted.  This suggests that differences in training choices may play a critical role in long-context processing, pointing to an interesting direction for future research. We have incorporated this analysis and discussion into Section 3.1.
>
>
> > ### [Q6] (Not a requirement) I wonder if there are plans to release the training set (e.g., gym-like setup) to facilitate reproducibility and further research.
>
> Thank you for this suggestion. As our work introduces a benchmark for evaluation, we do not curate a separate training set.
>
> However, to ensure full reproducibility and to actively facilitate further research, we are releasing our complete data generation pipeline and code. We have updated the submission with a zip file containing the dataset, all evaluation code, and the full implementation to regenerate the benchmark. Upon publication, this will be made publicly available. Since the core data curation pipeline is automated, our released resources can serve as a foundation for building gym-like training environments or for adapting the benchmark to new setups, enabling the community to extend this line of work.

---

### Official Review · Reviewer_ehn3 · 2025-11-01

**Soundness:** 2
**Presentation:** 3
**Contribution:** 2
**Rating:** 4
**Confidence:** 3

**Summary:**

This paper presents FronTalk, a conversational benchmark for frontend coding. At each turn, an LLM-based user simulator converts static intents into context-aware instructions that prompt the agent to edit the codebase. A web agent evaluates the developed website on two axes: instruction following, measured by test cases, and usability, measured by simulating a first-time user. Experiments on the new dataset reveal a common issue of forgetting: repeated edits to the same code region across turns often introduce regressions. To address this, the authors use an external web agent as a critic model that verifies compliance with instructions on the rendered site and produces textual critiques of failures. The approach substantially reduces forgetting.

**Strengths:**

1. The proposed task of conversational frontend coding is both novel and realistic. It is potentially important for real-world human-machine collaborative web development.

2. It is an interesting and valuable finding that existing LLM agents often break functionality introduced in earlier turns.

3. The proposed method, ACECODER, is simple yet effective.

**Weaknesses:**

1. The evaluation does not penalize the agent for adding unrequested functions or layout. Such additions may reduce usability, but the score does not reliably reflect that.

2. The evaluation relies mainly on an automatic agent, which is itself a complex problem, so the approach needs rigorous justification that is missing. The authors report a human study with 82.0 accuracy and a Cohen’s kappa of 62.7. It is unclear whether these are sufficient to show that the proposed metrics are a reliable proxy. The paper also omits key details of the human study, including the number of annotators, how they were recruited, any qualification screening, and the instructions they received. Similar issues exist for the LLM-based user simulator.

3. The method introduces an additional critic model that increases inference latency, yet the evaluation does not account for this cost.

4. In real-world web development, users often provide ambiguous requests and iterate. For example, they may ask to add a button and later decide to remove it. FronTalk does not consider these scenarios.

**Questions:**

Questions:

See weaknesses.

Others:

* Fig. 4: Please adjust the margin between the labels on the x-axis.

---

> ### Author Response · Authors · 2025-12-03
>
> > ### [W1] The evaluation does not penalize the agent for adding unrequested functions or layout.
>
> We agree that, in a strict sense, any feature not explicitly requested could be interpreted as a deviation from the specification. In real-world front-end development, however, **implicit requirements**, such as input validation functionalities for an input form, are both common and expected. An LLM that can correctly infer and implement these implicit expectations improves user experience, and therefore should not be penalized in evaluation.
>
> To address the concern of truly *unwanted* features that are **neither entailed nor implied by user instructions**, we conducted an additional study using GPT-5-mini to analyze outputs from three representative models. We identified an interesting pattern: **the prevalence of unwanted features correlates with model capability**. More capable models like GPT-4o and Qwen2.5-VL-72B introduce unwanted features very rarely, forming a small and non-dominant subset of errors. In contrast, less capable models like Qwen2.5-VL-7B generate unwanted features far more frequently, largely due to misinterpretations of the user request and incorrect implementations.
>
> | | % Website with unwanted features |
> |---|---|
> | GPT-4o | 12 |
> | Qwen2.5-VL-72B | 15 |
> | Qwen2.5-VL-7B | 49 |
>
> While we are unable to perform this evaluation across all model outputs due to resource constraints, we have included the evaluation script in the supplementary materials. This enables the community to further investigate this phenomenon and to better understand how model capabilities relate to the generation of unwanted content.
>
> > ### [W2] The evaluation relying on an automatic agent needs rigorous justification.
>
> In the original draft (Section 3.2), we validate the LLM-based evaluation pipeline by measuring its agreement with an internal human annotator. We observe an accuracy of 82.0 (Cohen’s $\kappa$ of 62.7) for pass rate and 84.0 (Cohen’s $\kappa$ of 66.7) for usability, indicating **substantial agreement** based on commonly accepted interpretations of the Cohen’s $\kappa$ metric [1].
>
> To further validate the reliability of LLM-based evaluation, we conducted two additional sets of experiments: expanded human evaluation and evaluation with multiple LLMs.
>
> **Expanded human evaluation**: We engaged four additional annotators, bringing the total to five. All annotators are computer science students with at least an undergraduate-level education. Agreement between LLM evaluation and aggregated human judgments yields an accuracy of 80.2 (Cohen’s $\kappa$ of 58.9) for pass rate and 69.3 (Cohen’s $\kappa$ of 56.3) for usability, indicating **moderate to substantial agreement**. The slightly lower agreement for usability stems from the inherently subjective nature of the metric. However, usability remains a crucial supplementary metric: subjective factors such as user experience significantly shape a front-end coder’s real-world utility, but have been largely overlooked in prior work. Our study is among the first to rigorously measure this dimension. Our work is among the first to explore measuring this dimension. Detailed setups are updated in Section 3.3 in the revision.
>
> **Evaluation with multiple LLMs**: We employ two additional models, Gemini-2.5-Pro and Qwen2.5-VL-72B, to evaluate pass rate and usability for four sets of outputs. The results are summarized below.
>
> | Evaluator$\to$ | GPT-4o | | Gemini-2.5-Pro | | Qwen2.5-VL-72B | |
> |---|---|---|---|---|---|---|
> | Models to be evaluated$\downarrow$ | **Pass rate** |**Usability** |**Pass rate** |**Usability** | **Pass rate** |**Usability** |
> | GPT-4o | 52.7 (rank=2) | 55.0 (rank=2) | 42.8 (rank=2) | 45.5 (rank=2) | 46.5 (rank=2) | 61.0 (rank=2) |
> | Gemini-2.5-Pro | 70.1 (rank=1) | 71.0 (rank=1) | 61.7 (rank=1) | 75.0 (rank=1) | 56.1 (rank=1) | 61.3 (rank=1) |
> | Qwen2.5-VL-7B | 19.7 (rank=4) | 26.8 (rank=4) | 16.6 (rank=4) | 13.3 (rank=4) | 20.6 (rank=4) | 38.5 (rank=4) |
> | Qwen2.5-VL-72B | 45.1 (rank=3) | 46.5 (rank=3) | 34.5 (rank=3) | 35.3 (rank=3) | 38.0 (rank=3) | 49.5 (rank=3) |
>
> While the absolute scores varied slightly across judges, **the relative ranking of all models remains consistent across all three LLMs**, demonstrating robust agreement in model ranking and evaluation. We additionally calculated the IAA metrics for pass rate evaluation, achieving an accuracy of 81.4 (Cohen’s $\kappa$ of 62.3), indicating **substantial agreement**. These results further demonstrate the reliability of our LLM-based evaluation framework. Detailed setups are updated in Appendix E.1.

---

> ### Author Response · Authors · 2025-12-03
>
> > ### [W3] The additional critic model in AceCoder method increases inference latency and cost.
>
> We would like to begin by clarifying that **our primary contribution is the benchmark itself** — the dataset and evaluation framework — which reveals the unique challenges of front-end code generation in multi-turn multi-modal settings. On this basis, AceCoder is presented as a **proof-of-concept** baseline to explore agent-based self-critique. While AceCoder demonstrates that this mechanism can effectively improve the performance, its current implementation is straightforward and not yet optimized, which indeed results in higher cost and latency.
>
> To better quantify the computational overhead, we measure AceCoder’s inference cost using GPT-4o on a subset of 10 dialogues. As in the table, the majority of the overhead comes from agent-based critique. **This largely reflects the broader efficiency limitations of current web agents**, which involve multi-turn interactions with the environment, a high-complexity observation space, and long multi-modal histories. These challenges are well known in existing systems and are not specific to our implementation.
>
> We acknowledge that optimizing efficiency is important, and we view it as an important next step for future work. Potential directions including web agent efficiency, history compression, and improved planning algorithms. We have added this analysis and discussion to Appendix C.2.
>
> | # LLM requests | # Input tokens | # Cached input tokens | # Output tokens | Est. cost |
> |---|---|---|---|---|
> | Vanilla inference | 100 | 14.4k | 728k | 205k | $3.00 |
> | AceCoder | 4.5k | 3.1M | 15.1M | 967k | $36.22 |
> | > Website candidate generation | 100 | 14.7k | 928k | 303k | $4.22 |
> | > Agent-based critique | 4.3k | 2.9M | 13.0M | 380k | $27.29
> | > Refined website generation | 90 | 148k | 1.2M | 284k | $4.7 |
>
> > ### [W4] In real-world web development, users often provide ambiguous requests and iterate. FronTalk does not consider these scenarios.
>
> To address this application scenario, we employ LLMs to simulate two types of *ambiguous users*: (1) **clarification-only user** that responds to direct questions but does not provide unsolicited details, and (2) **preference-only user** that verbalizes only minimal requirements but instead reveals intent through choices among model-generated options. These simulated users have full access to the true intent but are constrained to communicate in the specified style. All settings share the same interaction budget (10 turns) to ensure fairness.
>
> As shown below, both user types significantly degrade performance compared to standard multi-turn or single-turn settings. This highlights models' limited capability to actively elicit and infer user intent, directly affecting the task completion pass rate and the usability of generated websites. The flexibility to simulate new user types further highlights the benefits of LLM-based user simulation, as it can dynamically adapt to new types of users or interaction dynamics and thus further expand our benchmark. We have added the results and discussion in Section 3.2.
>
> | Turn | User type | Pass rate | Usability |
> |---|---|---|---|
> | Single | - | 51.4 | 50.0 |
> | Multi | Standard | 56.0 | 55.0 |
> | Multi | Clarification-Only | 23.9 | 31.0 |
> | Multi | Preference-Only | 16.4 | 21.0 |
>
> > ### [Q1] Fig. 4: Please adjust the margin between the labels on the x-axis.
>
> Thank you for pointing this out. We have adjusted Figure 4 (Figure 6 in the revision) to make it clearer.
>
> ---
>
> [1] Landis, J. Richard, and Gary G. Koch. "The measurement of observer agreement for categorical data." biometrics (1977): 159-174.

---

### Official Review · Reviewer_ENsF · 2025-11-02

**Soundness:** 3
**Presentation:** 4
**Contribution:** 3
**Rating:** 6
**Confidence:** 3

**Summary:**

This paper presents FronTalk, a new benchmark designed to evaluate code generation models in the domain of front-end web development across multiple conversational turns, incorporating both text and image feedback, and simulating realistic user interactions during website building.
The benchmark comprises a dataset of 100 dialogues (approximately 1,000 turns and 3.6k test cases) derived from real websites, paired with a multi-modal user simulator that provides both textual and visual cues. The paper introduces an agent-based evaluation framework to assess both code correctness and usability.
To address the "forgetting issue" where previously implemented features are overwritten during subsequent turns, they propose AceCoder, a baseline method employing agent-based critique as a mitigation strategy.

**Strengths:**

- This work addresses incorporating multi-modal (text & image) feedback into multi-turn code generation, specifically targeting front-end development, which is underexplored in current benchmarks.
- The dataset contains 1,000 conversational turns across 100 dialogues and 3,676 manually refined test cases for robust model evaluation. The data is grounded in real-world websites from diverse domains, thereby increasing the benchmark's practicality and relevance.
- Evaluations are comprehensive and cover a wide range of 14 different models.

**Weaknesses:**

- The dataset is generated using an LLM-based user simulator that generates context-aware instructions conditioned on prior dialogue. For evaluation, first-time users simulated by LLMs interact with each website, and a secondary LLM then compares the resulting trajectories and judges which interface is more usable. In this work, both dataset generation and evaluation rely heavily on LLMs, which can cause multiple problems: the evaluation is not reliable, the user simulator might not generate the follow-ups that are representative of actual multi-turn conversations, and the model is more likely to ask questions that are already in the model's distribution.

- Proposed Acecoder is inefficient, has high cost and latency, and uses the same LLM for evaluation and generation, which can cause bias.
The paper would be more academically sound if it were to present the dataset itself.

**Questions:**

Lots of hyperparameters have not been mentioned in the paper; please consider adding a section to discuss them in more detail.

---

> ### Author Response · Authors · 2025-12-03
>
> Thank you for your thoughtful review and for acknowledging the strengths of our work, including the novelty of our benchmark design, the quality of our real-world grounded dataset, and the comprehensiveness of our empirical evaluation.
>
> We address your questions and concerns below:
>
> > ### [W1] Evaluation relies heavily on LLMs, so the evaluation may not be reliable.
>
> In the original draft (Section 3.2), we validate the LLM-based evaluation pipeline by measuring its agreement with an internal human annotator. We observe an accuracy of 82.0 (Cohen’s $\kappa$ of 62.7) for pass rate and 84.0 (Cohen’s $\kappa$ of 66.7) for usability, indicating **substantial agreement** based on commonly accepted interpretations of the Cohen’s $\kappa$ metric [1].
>
> To further validate the reliability of LLM-based evaluation, we conducted two additional sets of experiments: expanded human evaluation and evaluation with multiple LLMs.
>
> **Expanded human evaluation**: We engaged four additional annotators, bringing the total to five. All annotators are computer science students with at least an undergraduate-level education. Agreement between LLM evaluation and aggregated human judgments yields an accuracy of 80.2 (Cohen’s $\kappa$ of 58.9) for pass rate and 69.3 (Cohen’s $\kappa$ of 56.3) for usability, indicating **moderate to substantial agreement**. The slightly lower agreement for usability stems from the inherently subjective nature of the metric. However, usability remains a crucial supplementary metric: subjective factors such as user experience significantly shape a front-end coder’s real-world utility, but have been largely overlooked in prior work. Our study is among the first to rigorously measure this dimension. Our work is among the first to explore measuring this dimension. Detailed setups are updated in Section 3.3 in the revision.
>
> **Evaluation with multiple LLMs**: We employ two additional models, Gemini-2.5-Pro and Qwen2.5-VL-72B, to evaluate pass rate and usability for four sets of outputs. The results are summarized below.
>
> | Evaluator$\to$ | GPT-4o | | Gemini-2.5-Pro | | Qwen2.5-VL-72B | |
> |---|---|---|---|---|---|---|
> | Models to be evaluated$\downarrow$ | **Pass rate** |**Usability** |**Pass rate** |**Usability** | **Pass rate** |**Usability** |
> | GPT-4o | 52.7 (rank=2) | 55.0 (rank=2) | 42.8 (rank=2) | 45.5 (rank=2) | 46.5 (rank=2) | 61.0 (rank=2) |
> | Gemini-2.5-Pro | 70.1 (rank=1) | 71.0 (rank=1) | 61.7 (rank=1) | 75.0 (rank=1) | 56.1 (rank=1) | 61.3 (rank=1) |
> | Qwen2.5-VL-7B | 19.7 (rank=4) | 26.8 (rank=4) | 16.6 (rank=4) | 13.3 (rank=4) | 20.6 (rank=4) | 38.5 (rank=4) |
> | Qwen2.5-VL-72B | 45.1 (rank=3) | 46.5 (rank=3) | 34.5 (rank=3) | 35.3 (rank=3) | 38.0 (rank=3) | 49.5 (rank=3) |
>
> While the absolute scores varied slightly across judges, **the relative ranking of all models remains consistent across all three LLMs**, demonstrating robust agreement in model ranking and evaluation. We additionally calculated the IAA metrics for pass rate evaluation, achieving an accuracy of 81.4 (Cohen’s $\kappa$ of 62.3), indicating **substantial agreement**. These results further demonstrate the reliability of our LLM-based evaluation framework. Detailed setups are updated in Appendix E.1.
>
> > ### [W1] Dataset generation relies heavily on LLMs, so that the user simulator might not generate the follow-ups that are representative of actual multi-turn conversations, and the model is more likely to ask questions that are already in the model's distribution.
>
> We acknowledge the concern that LLM-based user simulation may not perfectly capture human conversational dynamics. However, **our dataset generation methodology explicitly addresses this by grounding the data in real-world websites from diverse domains** — a strength the reviewer also highlights. During data curation, the core user intents are derived from a diverse collection of real-world websites, ensuring that the fundamental tasks reflect real user needs. In the user simulation stage, the LLM is **tightly constrained**: it is tasked only with contextualizing these pre-defined intents within the dialogue history, with minimal free-form generation. This ensures that the final instructions remain representative of the real distribution of webpage requirements. As reported in Section 2.3, the textual user simulator preserves 98% of the original intent. The visual user simulator achieves slightly lower intent preservation (76%) due to current limitations in tool-calling capabilities, but we expect this gap to narrow as LLMs capabilities continue to improve.
>
> Regarding the concern that LLMs may tend to generate instructions within their training data distribution, empirical evidence suggests this does not undermine the benchmark's difficulty: even the LLM used to generate the data (GPT-4o) achieves only a ~50% pass rate on the final benchmark. This demonstrates that **the benchmark presents a substantial and robust challenge despite its synthetic origins**.

---

> ### Author Response · Authors · 2025-12-03
>
> > ### [W2] Proposed Acecoder is inefficient, has high cost and latency.
>
> We would like to begin by clarifying that **our primary contribution is the benchmark itself** — the dataset and evaluation framework — which reveals the unique challenges of front-end code generation in multi-turn multi-modal settings. On this basis, AceCoder is presented as a **proof-of-concept** baseline to explore agent-based self-critique. While AceCoder demonstrates that this mechanism can effectively improve the performance, its current implementation is straightforward and not yet optimized, which indeed results in higher cost and latency.
>
> To better quantify the computational overhead, we measure AceCoder’s inference cost using GPT-4o on a subset of 10 dialogues. As in the table, the majority of the overhead comes from agent-based critique. **This largely reflects the broader efficiency limitations of current web agents**, which involve multi-turn interactions with the environment, a high-complexity observation space, and long multi-modal histories. These challenges are well known in existing systems and are not specific to our implementation.
>
> We acknowledge that optimizing efficiency is important, and we view it as an important next step for future work. Potential directions including web agent efficiency, history compression, and improved planning algorithms. We have added this analysis and discussion to Appendix C.2.
>
> | # LLM requests | # Input tokens | # Cached input tokens | # Output tokens | Est. cost |
> |---|---|---|---|---|
> | Vanilla inference | 100 | 14.4k | 728k | 205k | $3.00 |
> | AceCoder | 4.5k | 3.1M | 15.1M | 967k | $36.22 |
> | > Website candidate generation | 100 | 14.7k | 928k | 303k | $4.22 |
> | > Agent-based critique | 4.3k | 2.9M | 13.0M | 380k | $27.29
> | > Refined website generation | 90 | 148k | 1.2M | 284k | $4.7 |
>
> > ### [W2] Acecoder uses the same LLM for evaluation and generation, which can cause bias.
>
> For clarification, **AceCoder does not use the same LLM as the final evaluator**. AceCoder is a model-agnostic, add-on algorithm where the *base* LLM (e.g., GPT-4o, Qwen2.5-VL) powers a critique agent. This is separate from the *evaluator* LLM (GPT-4o in our experiments) that assesses final output against hidden test cases. Moreover, during the critique stage, the agent only has access to the user instructions and the current code — **it never accesses the hidden test cases**, eliminating risks of label leakage. We apologize for any lack of clarity in the original draft; we have revised Section 4 to make this clearer.
>
> > ### [Q1] Lots of hyperparameters have not been mentioned in the paper
>
> Thank you for this suggestion. We have significantly reorganized and expanded the appendix to include comprehensive experimental details. Key hyperparameters, including model configurations and inference settings, have been updated in Appendix C.1. We have also included the source code as supplementary materials. We believe these additions enhance the clarity and reproducibility.
>
> ---
>
> [1] Landis, J. Richard, and Gary G. Koch. "The measurement of observer agreement for categorical data." biometrics (1977): 159-174.

---

### Author Response · Authors · 2025-12-03
**General Response**

We sincerely thank all reviewers for their insightful feedback. We are delighted that the reviewers recognize the value of our FronTalk benchmark, highlighting its novelty (ENsF), realism (ENsF, ehn3), and potential importance for human-AI collaborative front-end development (ehn3, VZoF). We also appreciate the acknowledgment of our extensive experimental evaluation (ENsF, SnEG) and analysis (ehn3, SnEG), particularly our study of the forgetting issue (ehn3). Finally, we are grateful for the positive remarks on our proposed AceCoder method for its simple yet effective design for mitigating the forgetting issue (ehn3, SnEG).

The reviewers also raised several important concerns, which we have carefully addressed in our responses and revisions. The key shared concerns and our corresponding actions are summarized below:
1. **Evaluation reliability** (ENsF, ehn3, SnEG): Reviewers expressed concerns about the reliability of LLM agent-based evaluation. To address this, we perform **expanded human evaluation** with a total of five annotators to verify agreement with the LLM judge. We also conducted a **multi-LLM judge evaluation** using three distinct LLMs as evaluators, showing consistent model rankings and strong cross-LLM agreement. The corresponding revisions are in Section 3.3 and Appendix E.1.
2. **Cost and efficiency** (ENsF, ehn3, VZoF): Reviewers asked about computational costs, including general inference and evaluation overhead as well as the additional cost brought by AceCoder. We added a detailed cost breakdown for all components. Our results show that **the most computationally intensive components are those involving multi-turn agents**, including web-agent-based evaluation, tool-calling visual user simulation, and AceCoder’s agent-based critique. The revisions are in Appendix C.2.
3.  **Reproducibility and public release** (ENsF, ehn3, VZoF): Reviewers raised concerns about reproducibility and benchmark availability. To address this, we updated the appendix with comprehensive hyperparameter settings and implementation details. We also submit the full dataset, inference and evaluation code, and the data-generation pipeline as supplementary materials, and we will make them publicly available upon publication.

Beyond these shared concerns, we also address the individual comments from each reviewer in the corresponding responses and revisions.

In our revision, we highlight all critical updates in blue. The main updates are summarized as follows:
1. **Additional models evaluated**: We evaluate 8 additional models, bringing the total number of models from 12 to 20. Results are updated in Table 2.
2. **Additional analysis**, including: long-context degradation analysis in Section 3.1; simulation of ambiguous users in Section 3.2; expanded human evaluation in Section 3.3; cost analysis in Appendix C.2; cross-LLM agreement analysis in Appendix E.1; and confidence interval analysis in Appendix E.2.
3. **Expanded related-work comparison**: expanded discussion in Section 5; and additional comparative examples in Figure 8.
4. **Revisions for clarity**: We reorganized the appendix and added a table of contents, moved the introduction of the single-turn baseline to Section 2.1, and added clarification of AceCoder in Section 4.

We apologize for the delayed response as we worked on the additional experiments and human evaluations. We feel sorry that we are not able to engage in any further discussions. We hope our responses and revisions address the reviewers’ concerns and strengthen the contribution of our work.

---

### Meta-Review · Area_Chair_exLq · 2026-01-07

**Summary:**

The most important reviewers' concerns are the following:
1. Rev ENsF: Both dataset generation and evaluation rely heavily on LLMs, which can bias the dataset.
2. Rev. ENsF and ehn3: the user simulator might not generate representative multi-turn conversations, and the model is more likely to ask questions that are already in the model's distribution.
3. Rev. ENsF, ehn3 and SnEG: Not clear if the human evaluation agreement is enough, especially with a single annotator.
4. Rev. ehn3: Score does not consider the introduction of unwanted features such as functions or layouts.
5. Rev. SnEG: What is the difference between the proposed multi-turn benchmark and WebGen-Agent?

**Reviewer Concerns:**

The provided answers from the authors:
1. A human annotator validates the LLM-based evaluation and has substantial agreement with the model.
2. Although built on LLM, the dialogs are based on real websites with real user intents and needs. Finally, the dataset seems challenging, which is a sign that the dialogs are not easy to foresee.
3. The human evaluation was also expanded to a total of 5 human annotators and added two extra LLMs, Gemini and Qwen, for evaluation, showing substantial agreement.
4. The authors recognised the problem and performed an additional study checking the percentage of unwanted features and showing that unwanted features are introduced, especially on small models.
5. The main difference is that FronRalk is a multi-turn conversional benchmark with evolving user requests, while WebGen-Agent is multi-turn, but with a fixed initial request.

The answers provided to Q1, Q3, and Q5 were satisfactory.  For Q2, the provided answer is not fully satisfactory, as a challenging dataset does not mean that the multi-turn conversations are diverse and similar enough to real situations.
For Q4, the additional evaluation shows that the models have problems with unwanted features and the proposed metrics do not account for that.
These two issues are quite serious for a benchmark paper, as it does not provide guarantees that the synthetic evaluation data is a good representative of a real distribution and the evaluation does not take into account possible issues with extra functions or layouts added by the model during the multi-turn code generation. Thus, I consider that this submission should be improved on these issues before acceptance.

**Reviewer Scores:**

Rev. ENsF: 6 -> 6
The authors provided satisfactory answers to most of the questions, so I think rev. would keep their score.
Rev. ehn3: 4 -> 2
With an additional experiment, the authors showed that agents could introduce unrequested functions, especially for small models, and this is not taken into account in the evaluation. This is an important issue, in my opinion. In addition, FronTalk does not consider ambiguous requests, as it is a synthetic dataset generated by LLMs. Therefore, I think that rev. would consider a negative score.
Rev. SnEG: 4 -> 6
The difference with WebGen-Agent was clarified, as well as the human validation of the LLM-based evaluation. Also, the other answers were meaningful; therefore, I believe rev. could increase their score to 6.
Rev. VZoF: 4
This review is too short and does not consider any important issues of the paper. Thus, it won't be considered.

---

### Decision · Program_Chairs · 2026-01-26

Reject